# AUTOLRS: AUTOMATIC LEARNING-RATE SCHEDULE BY BAYESIAN OPTIMIZATION ON THE FLY

**Yuchen Jin, Tianyi Zhou, Liangyu Zhao**
University of Washington
`{yuchenj, tianyizh, liangyu}@cs.washington.edu`

**Yibo Zhu, Chuanxiong Guo**
ByteDance Inc.
`{zhuyibo, guochuanxiong}@bytedance.com`

**Marco Canini**
KAUST
`marco@kaust.edu.sa`

**Arvind Krishnamurthy**
University of Washington
`arvind@cs.washington.edu`

## ABSTRACT

The learning rate (LR) schedule is one of the most important hyper-parameters needing careful tuning in training DNNs. However, it is also one of the least automated parts of machine learning systems and usually costs significant manual effort and computing. Though there are pre-defined LR schedules and optimizers with adaptive LR, they introduce new hyperparameters that need to be tuned separately for different tasks/datasets. In this paper, we consider the question: *Can we automatically tune the LR over the course of training without human involvement?* We propose an efficient method, `AutoLRS`, which automatically optimizes the LR for each training stage by modeling training dynamics. `AutoLRS` aims to find an LR applied to every $\tau$ steps that minimizes the resulted validation loss. We solve this black-box optimization on the fly by Bayesian optimization (BO). However, collecting training instances for BO requires a system to evaluate each LR queried by BO's acquisition function for $\tau$ steps, which is prohibitively expensive in practice. Instead, we apply each candidate LR for only $\tau' \ll \tau$ steps and train an exponential model to predict the validation loss after $\tau$ steps. This mutual-training process between BO and the loss-prediction model allows us to limit the training steps invested in the BO search. We demonstrate the advantages and the generality of `AutoLRS` through extensive experiments of training DNNs for tasks from diverse domains using different optimizers. The LR schedules auto-generated by `AutoLRS` lead to a speedup of $1.22\times$, $1.43\times$, and $1.5\times$ when training ResNet-50, Transformer, and BERT, respectively, compared to the LR schedules in their original papers, and an average speedup of $1.31\times$ over state-of-the-art heavily-tuned LR schedules.

## 1 INTRODUCTION

In the regime of deep learning, the success of training largely depends on the choice of the learning rate (LR) schedule, since most optimizers will have difficulty traversing a non-smooth and non-convex loss landscape with multiple local minimums and possibly saddle points (Kawaguchi, 2016; Jin et al., 2017; Goodfellow et al., 2016; Li et al., 2018a). To achieve stable and fast convergence towards a solution with good generalization performance, one has to tune the LR schedules carefully for different tasks (Nar & Sastry, 2018; Jastrzębski et al., 2017). This tuning is usually non-trivial and requires many trial-and-error iterations that are computationally expensive. Moreover, the randomness of the widely-used mini-batch stochastic gradient descent (SGD) may introduce more uncertainty and difficulty in the tuning process. For the same reasons, it is also hard to directly formulate the search of the LR schedule as a well-posed optimization problem and address it through standard optimization.

The broadly-adopted strategy is to either pick one from a family of pre-defined LR schedules or apply an optimizer that has a built-in mechanism changing the LR adaptively. However, we have a limited number of choices for pre-defined LR schedules, most of which are simple functions such as exponent or cosine and thus cannot perfectly align with the non-smooth loss landscape. The latter set of adaptive optimizers, e.g., Adam (Kingma & Ba, 2015) and Adadelta (Zeiler, 2012), are extended from convex optimization and rely on strong assumptions to make the convergence properties hold. Moreover, the methods in both categories introduce new hyper-parameters that have to be tuned separately for different tasks or datasets, requiring significant human involvement.

In this paper, we study the question: *can we automatically tune the LR over the course of training without human involvement?* At the beginning of every $\tau$ steps (i.e., a "stage" in our method), we seek to identify an LR that optimizes the validation loss (i.e., an empirical estimate of the generalization error) at the end of the stage. To do so, we employ Bayesian optimization (BO) that treats the validation loss as a black-box function of LR. BO simultaneously updates a posterior estimation of the black-box function and searches for the best LR with respect to the posterior. This approach is, however, computationally expensive since estimating the posterior needs many (input, output) instances of the function, and acquiring each instance costs $\tau$ steps of training. We, therefore, develop a simple yet efficient approximation: for every LR that BO decides to evaluate, we train the model by using the LR for only $\tau' \ll \tau$ steps and use the validation loss over the $\tau'$ steps to train a time-series forecasting model that provides a prediction of the validation loss after $\tau$ steps. As we will show later, an exponential model suffices to produce accurate predictions when using a small $\tau' = \tau/10$. Then, `AutoLRS` can allow BO to explore ten different LRs in each stage and still bound the total running time to approximately twice the training cost associated with the generated schedule, i.e., the time spent to find the stage-specific LRs is roughly equal to the time spent training the model with the identified LRs.

`AutoLRS` does not depend on a pre-defined LR schedule, dataset, or a specified task and is compatible with almost all optimizers. Hence, it can be generally deployed across a broad range of ML tasks without much human involvement or expensive tuning over choices of LR schedules and their hyperparameters. Moreover, since it directly minimizes the validation loss, it does not only accelerate the convergence but also improves the generalization compared to just minimizing the training loss. Furthermore, `AutoLRS` only needs to update two extremely light-weight models, i.e., the BO posterior and the exponential forecasting model, and it is efficient in exploring the loss landscape. Hence, it does not result in notable extra costs in either memory or computation. Note that `AutoLRS` searches for better LRs based on the training dynamics, which can be seen as a form of self-supervision. The interaction between BO and the forecasting model is an example of mutual learning, where one produces training data for the other.

In experiments, we apply `AutoLRS` to train three representative DNNs widely used in practice, i.e., ResNet-50 (He et al., 2016a) on ImageNet classification (Russakovsky et al., 2015); Transformer (Vaswani et al., 2017) and BERT (Devlin et al., 2019) for NLP tasks. Though they have been extensively studied and have hand-tuned LR schedules, the LR schedules computed by `AutoLRS` are faster than the original, hand-tuned, LR schedules by $1.22\times$, $1.43\times$, and $1.5\times$ for training ResNet-50, Transformer, and BERT, respectively, in terms of the training steps used to update the DNN (i.e., excluding the costs of the LR/hyperparameter search). It meanwhile achieves test-set performance better or on par with state-of-the-art results. We also carefully hand-tuned two state-of-the-art learning rate schedules, CLR (Smith, 2017) and SGDR (Loshchilov & Hutter, 2017), and conducted more than ten experiments with different CLR/SGDR hyperparameters on each model. `AutoLRS` still has an average speedup of $1.29\times$ and $1.34\times$ across the three models, in terms of training steps, compared to the best CLR and SGDR LR schedules, respectively. The `AutoLRS` implementation is available at `https://github.com/YuchenJin/autolrs`.

## 2 RELATED WORK

**Learning rate scheduling:** In contrast to traditional LR schedules with a monotone decreasing sequence of LRs and multi-step LR schedule, a recent class of LR schedules propose to apply multiple cycles of LR decay. Cyclical Learning Rate (CLR) changes LR from a maximal LR ($\eta_{\max}$) to a minimal LR ($\eta_{\min}$) at a pre-defined frequency and achieves faster convergence for some DNNs (Smith, 2017). The approach requires a "*LR range test*" to estimate the minimal and maximal LR. The *LR range test* trains the model with a linearly-increasing LR between a low LR

and a high LR, and finds the LR range ($[\eta_{\min}, \eta_{\max}]$) over which the training loss decreases. The authors proposed three variants of CLR: *triangular2* that halves the maximum LR bound after each cycle; *exp_range* that exponentially reduces the maximum LR bound after each cycle; and *1cycle* containing only one triangular cycle (Smith, 2018). Similar to CLR, Stochastic Gradient Descent with Warm Restarts (SGDR) restarts the LR and then applies cosine annealing/decay at a pre-defined frequency (Loshchilov & Hutter, 2017). Neither CLR or SGDR is automatic, because they are quite sensitive to their hyperparameters, which require careful hand-tuning. CLR and SGDR may even cause undesirable divergence in loss during training with suboptimal hyperparameters (see §5).

**Learning rate adaptation with hypergradient descent:** Aiming for the same goal of automatically tuning the LR, the hypergradient based technique (Almeida et al., 1998; Franceschi et al., 2017; Baydin et al., 2018; Donini et al., 2020) optimizes the LR schedule by applying gradient descent of the objective function w.r.t. the LR during training. In addition to the initial value of the regular LR, it introduces an additional hypergradient LR whose initial value is another hyperparameter to be specified. We experimentally show that this technique is subject to overfitting, it is quite sensitive to its two hyperparameters, and it is unable to match the state-of-the-art test-set performance on the models we test (§A.5.1). We also compare its performance against `AutoLRS` (§A.5.2).

**DNN hyperparameter optimization:** Automatic hyperparameter searching for DNNs has been broadly studied in recent years. When applied to learning rates, they can determine an optimized value for LR that is kept constant (or constrained to be a pre-defined shape) through the entire training process, as opposed to determining an LR schedule. They can be primarily categorized into Bayesian optimization based approaches (Hutter et al., 2011; Snoek et al., 2012; Bergstra et al., 2013), bandit-based solutions (Li et al., 2017; 2018b), hybrid approaches that combine bandit-based and Bayesian optimization based approaches (Falkner et al., 2018; Zela et al., 2018), and population-based methods (Jaderberg et al., 2017; Parker-Holder et al., 2020). It might be possible to extend these techniques to determine a LR schedule with an optimized LR for each training stage, but it is not sample-efficient and time-efficient to do so since the LR schedule would correspond to hundreds or thousands of hyperparameters.

**Optimization methods with adaptive LR:** These optimizers can adaptively adjust LR for each training step by maintaining an estimate of a better learning rate separately for each parameter in the DNN. Adagrad (Duchi et al., 2011) applies lower LRs to parameters with larger accumulated gradients and higher learning rates to the ones with smaller accumulated gradients. RMSprop (Tieleman & Hinton, 2012), AdaDelta (Zeiler, 2012), and Adam (Kingma & Ba, 2015) were later proposed to address the issue in Adagrad that the model stops learning due to the continual decay of LR. These optimizers with adaptive LR are orthogonal to our automatic LR scheduler, and they still require a global learning rate schedule, which can be obtained from our `AutoLRS`. In particular, their default hyperparameters do not always work well and need careful tuning, e.g., Adam's default LR 0.001 performs poorly in training BERT and Transformer, and a better-tuned LR schedule can significantly reduce the training time (§5). Recent optimization methods (Schaul et al., 2013; Mahsereci & Hennig, 2015) proposed to remove the need for LR tuning in SGD altogether, but they are not widely used potentially due to their limited applicability and sub-optimal performance (Baydin et al., 2018).

## 3 PROBLEM FORMULATION

Training of DNNs can be written in a general form of minimizing a loss function $L(x; \theta)$ over training samples $x \in D_{train}$, where $\theta$ represents the model weights being optimized. The minimization is conducted by applying an optimizer that updates $\theta$ iteratively. For example, at each step $t$, mini-batch SGD updates $\theta$ using the gradient computed on a mini-batch of samples $B_{train} \subseteq D_{train}$:

$$\theta_{t+1} = \theta_t - \frac{\eta_t}{|B_{train}|} \sum_{x \in B_{train}} \nabla_\theta L(x; \theta_t), \tag{1}$$

where $\eta_t$ is the learning rate (LR) at step $t$ and $\nabla_\theta L(x; \theta_t)$ denotes the gradient of the loss $L(x; \theta)$ w.r.t. $\theta_t$ at step $t$. Given $B_{train}$ and $\theta_t$, $\theta_{t+1}$ can be represented as a function of LR $\eta_t$, i.e., $\theta_{t+1}(\eta_t)$.

Our ultimate goal is to search for an optimal schedule of LR, i.e., a sequence of LRs $\eta_{1:T} \triangleq (\eta_1, \eta_2, \cdots, \eta_T)$ applied to the total $T$ training steps, such that the generalization error can be minimized. Ideally, we need to optimize the entire sequence of LRs. This, however, is intractable in practice given the large number of possible LR schedules and since evaluating each one of those possible LR schedules requires a full training of $T$ steps. Hence, we break down the LR schedule optimization into a dynamic optimization of a constant LR for every $\tau$ steps, which we refer to

as a "*training stage*". Since most tasks prefer a relatively small LR due to the non-smoothness of DNNs' loss landscapes, when $\tau$ is also small, the LR-resulted change on the validation loss might be too small and overwhelmed by the randomness of mini-batch SGD. Hence, in this case, we need to increase $\tau$, so the effect of LR $\eta$ on the validation loss can be accumulated for more steps to overcome noise. A large $\tau$ also reduces the frequency of applying LR search and saves computation. On the other hand, setting $\tau$ to be too large might lose some optimality of the induced LR schedule. Therefore, we need to trade-off the above two issues to find an appropriate $\tau$. In our final algorithm, we propose a curriculum for $\tau$, i.e., we start from a small $\tau$, in line with the greater volatility during early stages, and gradually increase $\tau$ as training proceeds (as described in §4.4). Since we mainly focus on LR search within a stage, for simplicity, we will use $\tau$ instead of $\tau_t$ for the exposition below.

We study a greedy approach and split the whole training process into multiple *stages* of $\tau$ steps each. We choose an LR at the beginning of each *stage* and apply $\tau$ steps of optimization using this LR, i.e., at step-$t = 0, \tau, 2\tau, \cdots, T - \tau$, we aim to find the LR $\eta_{t:t+\tau}$ that minimizes the validation loss on $D_{val}$ (i.e., an estimate of the generalization error) after step-$(t + \tau)$. This can be formulated as:

$$\min_\eta \sum_{x \in D_{val}} L(x; \theta_{t+\tau}(\eta)), \quad t = 0, \tau, 2\tau, \cdots, T - \tau. \tag{2}$$

We try to sequentially solve $\lfloor T/\tau \rfloor$ sub-problems of the above form. However, we cannot apply standard optimization to solve each sub-problem in practice because: $(i)$ it is a high-order optimization of $\eta$ since we need to unroll $\theta_{t+\tau}$ in Eq. (2) backward for $\tau$ steps using Eq. (1), which requires prohibitive memory and is unstable for DNNs; $(ii)$ one step of optimizing $\eta$ needs to apply $\tau$ steps of optimization on $\theta$, which is costly and weakens the advantage of searching LR for better efficiency. To avoid these issues, we treat the objective function in Eq. (2) for $t : t + \tau$ as a black-box function $f_t(\eta)$ and study how to optimize it based on the observed training dynamics through Bayesian optimization (BO).

# 4    AUTOMATIC LEARNING RATE SCHEDULE SEARCH

We first elaborate on the details of our BO algorithm (§4.1) that identifies the LR for each stage[1]. However, collecting even one data point $(\eta, f(\eta))$ for BO requires us to train the model for $\tau$ steps, which is costly and impractical since the LR computed by the entire BO process is used for only $\tau$ steps. To reduce the cost of generating instances of $(\eta, f(\eta))$, in §4.2 and §A.3, we propose to train a light-weight time-series forecasting model to predict $f(\eta)$ based on the validation loss observed during the first $\tau'$ ($\tau' \ll \tau$) steps of applying LR $\eta$. We find that a simple exponential model suffices to produce accurate predictions. Our LR search then reduces to a multi-training process between BO and the forecasting model, where one produces training instances for the other. The resulting algorithm can automatically find an LR schedule without introducing significant extra computation.

## 4.1    BAYESIAN OPTIMIZATION

BO (Shahriari et al., 2016) is one of the state-of-the-art techniques for black-box optimization. It applies exploration and exploitation to the objective by sequentially and actively querying the function values of some input instances. Specifically, BO uses Gaussian process as a *surrogate model* (prior) to fit the black-box objective function $f(\eta)$. It sequentially updates a posterior of $f(\eta)$ by using its likelihood on newly evaluated $(\eta_i', y_i = f(\eta_i') + \epsilon)$ pairs[2], where $y_i$ is a noisy observation of $f(\eta_i')$ and is the validation loss after $\tau$ steps. Then, it finds the next $\eta_{i+1}'$ to evaluate based on an acquisition function $u_i(\eta)$ defined by the posterior mean $\mu_i(\eta)$ and standard deviation $\sigma_i(\eta)$. $u_i(\eta)$ performs a trade-off between exploration (i.e., large $\sigma_i(\eta)$) and exploitation (i.e., small $\mu_i(\eta)$). In `AutoLRS`, we use Lower Confidence Bound (LCB) (Cox & John, 1992; Auer, 2002) as $u_i(\eta)$. Given $\eta_{1:i}'$ and their corresponding validation loss $y_{1:i}$, we determine the next LR $\eta_{i+1}$ by minimizing LCB, i.e.,

$$\eta_{i+1}' = \arg\min_\eta u_i(\eta), \quad u_i(\eta) \triangleq \mu_i(\eta) - \kappa\sigma_i(\eta), \tag{3}$$

where $\mu_i(\eta)$ and $\sigma_i(\eta)$ are defined in Eq. (7) in §A.1, $\kappa$ is a positive hyper-parameter to balance exploration and exploitation. In experiments, $\kappa = 1000$ works consistently well. BO repeats the above process until it achieves a precise posterior distribution of $f(\eta)$. See §A.1 for more details.

---

[1] Since this section mainly focuses to solve a sub-problem of Eq. (2) within one stage $t : t + \tau$, we temporarily remove the subscript $t$ from $f_t(\eta)$ and other variables/functions that do not change within a stage for simplicity.

[2] Here $i = 1, \cdots, k$ for $k$ steps in BO: it indexes the exploration step of BO within a training stage and differs from subscript $t$ indexing the training steps. We use superscript $'$ on $\eta'$ to mark the LRs explored by BO.

---

**Algorithm 1:** `AutoLRS`

---

**Input** : (1) Number of steps in each training stage, $\tau$
       (2) Learning-rate search interval $(\eta_{\min}, \eta_{\max})$
       (3) Number of LRs to evaluate by BO in each training stage, k
       (4) Number of training steps to evaluate each LR in BO, $\tau'$
       (5) Trade-off weight in the acquisition function of BO, $\kappa$

1 **while** *not converge* **do**
2     initialize a GP prior: $\mu_0(\eta) = 0, \sigma_0^2(\eta) = K(\eta, \eta)$ defined in Eq. (4) in §A.1;
3     $c \leftarrow$ checkpoint of model parameters and optimizer states;
4     **for** $i \leftarrow 1$ **to** k **do**               /* mutual-training loop between BO and loss forecasting model */
5         choose the next LR to explore: $\eta_i' = \arg\min_\eta \mu_{i-1}(\eta) - \kappa\sigma_{i-1}(\eta)$;
6         $y_{1:\tau'} \leftarrow$ train the DNN with LR $\eta_i'$ for $\tau'$ steps and record the corresponding validation loss series;
7         $y_n \leftarrow$ train an exponential forecasting model on $y_{1:\tau'}$ and predict the validation loss after $\tau$ steps;
8         update the GP posterior by $(\eta_i', y_i)$ and update new $\mu_i(\eta)$ and $\sigma_i(\eta)$ using Eq. (7) in §A.1;
9         restore the checkpoint $c$ of model parameters and optimizer states;
10     **end**
11     $\eta^* \leftarrow$ the LR with the minimal predicted validation loss $\mu_k(\eta)$ among the k explored LRs $\eta_{1:k}'$ above;
12     train the DNN using LR $\eta^*$ for $\tau$ steps;       /* training model using BO-searched best learning rate */
13 **end**

---

## 4.2 Time-series Forecasting Model of Loss

Typically, BO would requires $\tau$ training steps to measure the validation loss associated with every LR $\eta$ that it considers during a stage. This is computationally expensive. We now introduce a simple yet effective approach that substantially reduces the number of training steps required to evaluate each LR candidate: for each LR $\eta$ that is evaluated, we only apply it for $\tau' \ll \tau$ steps and use the validation loss observed in the $\tau'$ steps to train a short-term time-series forecasting model. We then use the resulting forecasting model to predict the validation loss after $\tau$ steps.

In numerous experiments, we observed that when a DNN is trained with a *reasonable* LR, the validation loss typically decreases exponentially and converges to a small value. We show examples of practical loss time series and their exponential-model fitting results in Figure 3. Moreover, recent deep learning theory (Allen-Zhu et al., 2019b) also proves the linear convergence of training DNNs. In addition, a simple model to fit the observed loss time-series can filter the noise and avoid possible overfitting. Hence, we propose to train an exponential model in the form of $L(t) = a\exp(bt) + c$ with parameters $a, c$ and $b < 0$ and for $t = 1, \ldots, \tau$, as the forecasting model for the time series of the validation loss in a training stage of $\tau$ steps with a given LR $\eta$. §A.2 describes how we estimate $a$, $b$, and $c$ based on the validation loss observed in the first $\tau'$ steps, and §A.3 describes how we filter out noise and outliers.

## 4.3 Mutual Training Between BO and Exponential Prediction

We present the complete procedure of `AutoLRS` in Algorithm 1. It sequentially optimizes LR for every training stage during the training of a DNN model, solely based on the observed training dynamics, and it can be seen as a form of self-supervision. For each training stage, it searches for the LR that leads to the largest improvement in the validation loss via an efficient black-box function optimization conducted by a mutual training loop between Bayesian optimization and a short-term forecasting model for each loss series. It then applies the best LR among the explored ones for $\tau$ steps and repeats the above process until convergence.

In line 5, the algorithm solves a constrained optimization problem over $\eta$, in the range of $[\eta_{min}, \eta_{max}]$. In practice, we prefer a large learning-rate search interval $(\eta_{\min}, \eta_{\max})$, across orders of magnitude, but also need fine-grained optimization over small LRs. Hence, we operate on $\eta$ in its log-scale space, i.e., we replace $\eta$ by $\log\eta$ in Algorithm 1, except in lines 6 and 12 when we use the original LR (rather than $\log\eta$) to train the DNN.

At the end of each iteration in the mutual training loop (line 9), we restore the checkpoint $c$ of model parameters and optimizer states to the one saved at the beginning of the training stage [3]. By doing so,

---

[3] We save the checkpoint in the CPU memory, so `AutoLRS` does not have GPU memory overhead.

we guarantee that the $k$ different LRs all start from the same model and their losses can be compared. §A.4 illustrates how BO learns the underlying function in practice for early and late stages of training.

**Hyperparameters:** `AutoLRS` substantially reduces the amount of hyperparameters that need to be hand-tuned in existing LR schedules or policies. However, as shown in Algorithm 1, we still have hyperparameters in `AutoLRS`. First, we need to set a search interval $(\eta_{\min}, \eta_{\max})$ for LR. However, this interval can be reasonably wide by using an *LR range test* (Loshchilov & Hutter, 2017) as we will show in §5. Secondly, our default settings of $k$, $\tau'$, $\tau$, and $\kappa$ work well for a diverse set of DNN models from different domains and tasks, though it is possible to achieve further improvements by fine-tuning them.

### 4.4 PRACTICAL IMPROVEMENTS

We found the following modifications can further improve the performance of `AutoLRS` in practice.

**Gradually increase $\tau$ over the course of training:** Often, in DNN training, the loss and the model parameters experience rapid changes only during the first few epochs before they enter a phase of stable improvement. Our approach can adapt to this phenomenon. For the early stages, when the loss is less predictable for the time-series forecasting model, we use a small $\tau$ (and $\tau'$). As training proceeds and the model becomes stable, we gradually increase $\tau$ (and $\tau'$) and adjust the LR more lazily. This curriculum of increasing $\tau$ places more exploration in earlier stages and more exploitation in later stages. In practice, we start with $\tau = 1000$ and $\tau' = 100$, and double them after every stage until it reaches $\tau_{\max}$. $\tau_{\max}$ is a hyperparameter that limits the maximum number of steps in a stage. We will discuss more of $\tau_{\max}$ in §5. This gradual increase of $\tau$ can provide stability to the LR schedule search. Similar strategies have been widely used in previous pre-defined LR schedules, e.g., the multi-stage schedule with increasing epochs within each stage, and some recent cyclical LR schedules (Loshchilov & Hutter, 2017).

**Minimizing training loss in early stages:** Computing the validation loss series for a candidate $\eta'$ requires considerable computation if we were to use the entire validation dataset at each step of mutual training. Recall, however, that the primary purpose of minimizing the validation loss instead of the training loss is to avoid overfitting on the training set when the training loss notoriously deviates from the generalization error. However, a variety of empirical evidence and recent theory (Allen-Zhu et al., 2019a) show that overfitting is unlikely while training over-parameterized DNNs due to the inductive bias of random initialization and SGD, especially during the early phase of training. Hence, in practice, for the first several training stages, we can safely approximate the validation loss in our method by the corresponding training loss, which is a by-product of forward propagation and free to obtain. In later stages (i.e., once $\tau$ reaches $\tau_{\max}$), since the model is stable and the loss changes smoothly, we can evaluate the validation loss on a small subset of the validation set without compromising robustness. In our experiments, this set is composed of merely 10 mini-batches, and we evaluate the validation loss on them every 50 training steps (as opposed to every step). Therefore, the evaluation of validation loss in our approach does not introduce notable extra computations[4].

## 5 EXPERIMENTS

We now evaluate `AutoLRS` by applying it to three widely-used and representative DNNs: ResNet-50, Transformer, and BERT. Here are some highlights:

- The LR schedules computed by `AutoLRS` are $1.22\times$, $1.43\times$, and $1.5\times$ faster, in terms of training steps, than the original, hand-tuned LR schedules for ResNet-50, Transformer, and BERT, respectively. Meanwhile, it improves or matches the test-set performance.
- For each model, we carefully hand-tuned CLR and SGDR using more than ten experiments with different CLR/SGDR hyperparameters. Across the three models, the LR schedules computed by `AutoLRS` achieve an average speedup of $1.29\times$ and $1.34\times$, in terms of training steps, over the best tuned LR schedules under CLR and SGDR, respectively. While CLR and SGDR had to be run

---

[4]The per-candidate-LR total cost of evaluating validation loss during the BO search in a later stage is $10 \cdot \Delta \cdot \lfloor \tau'/50 \rfloor$, where $\Delta$ is the time for computing the validation loss on one mini-batch. Since the cost of back propagation is roughly twice as that of forward propagation (Wen et al., 2018), the total cost of evaluating validation loss is approximately $1/15$ of the training time spent on BO search.

for at least 10 trials to find a good LR schedule, `AutoLRS` only costs slightly over $2\times$ the training time associated with the computed LR schedule even after accounting for the BO search cost.

- `AutoLRS` is robust to the change of hyperparameters and consistently finds better LR schedules than other baselines. In contrast, CLR and SGDR are sensitive to the choices of hyperparameters.
- We perform ablation studies in §A.5.4 to demonstrate that both BO and the exponential forecasting model are essential for `AutoLRS` to find good LR schedules.
- Hypergradient descent is subject to overfitting, and it is unable to match the state-of-the-art test-set performance using all the guideline values of its two hyperparameters on VGG-16 (Simonyan & Zisserman, 2015) and ResNet-50 (§A.5.1). In contrast, `AutoLRS` can consistently improve or match the state-of-the-art test-set performance with different $\tau_{\max}$ values using fewer training steps than the hand-tuned LR schedules (§A.5.2).
- Using Hyperband (Li et al., 2017) for LR schedule search incurs a high computational overhead. Moreover, it cannot find an LR schedule that matches the state-of-the-art accuracy (§A.5.3).

**Baseline Setup:** ML practitioners typically need to hand-tune the LR schedules carefully for a long time to achieve satisfying performance, so the LR schedule adopted in each model's original paper is a presumably tough-to-beat baseline to compare with. For CLR and SGDR, we hand-tune their hyperparameters separately for each DNN. Hyperparameters in CLR include the high/low LR for the *LR range test* to sweep, the number of steps to perform the test, the number of steps in each triangular cycle, and the choice of variants (*triangular2*, *exp_range*, *1cycle*) introduced in §2. Hyperparameters in SGDR include the number of steps/epochs in each cycle and the initial LR at the beginning of each cycle. We carefully tuned these hyperparameters separately for each DNN and chose the LR schedule producing the *best* validation-set performance among $\geq$10 trials of different hyperparameters.

**Hyperparameters in AutoLRS:** In our default setting, we set $k = 10$ and $\tau' = \tau/10$ so that the training steps spent on BO equals the training steps spent on updating the DNN model. We start from $\tau = 1000$ and $\tau' = 100$ and double $\tau$ and $\tau'$ after each *stage* until $\tau$ reaches $\tau_{\max}$. We use $\tau_{\max} = 8000$ for ResNet-50 and Transformer, $\tau_{\max} = 32000$ for BERT. We also tried $\tau_{\max} = 8000$, 16000, and 32000 for each DNN and found that the resulting LR schedules are not very sensitive to $\tau_{\max}$. (An analysis of the sensitivity to $\tau_{\max}$ is in §A.5.2.) The LR search interval $(\eta_{\min}, \eta_{\max})$ for ResNet-50, Transformer, and BERT are $(10^{-3}, 1)$, $(10^{-4}, 10^{-2})$, and $(10^{-6}, 10^{-3})$, respectively. These are easily found by an *LR range test* (Loshchilov & Hutter, 2017).

**ResNet-50:** ResNet (He et al., 2016a;b) is one of the most popular DNNs in computer vision tasks. We train ResNet-50 on ImageNet (Russakovsky et al., 2015) using SGD with momentum on 32 NVIDIA Tesla V100 GPUs with data parallelism and a mini-batch size of 1024. The LR schedule in the original paper adopts a *warmup* phase of 5 epochs at the beginning and performs a 3-step decay as in (Goyal et al., 2017). Figure 1a presents different LR schedules for training ResNet-50 on ImageNet. We report how their top-1 accuracy on the validation set[5] changes during training in Figure 1b. `AutoLRS` achieves a speedup of $1.19\times$ and $1.22\times$ over SGDR and the original LR schedule respectively but is slightly (i.e., 5.4%) slower than CLR. Note that the best CLR result is achieved after 10 trials of heavy hand-tuning to hyperparameters. (In fact, 7 out of 10 CLR trials failed to achieve the best possible test-set accuracy, and the second best and the third best trials are 5.4% and 7.9% slower than `AutoLRS`). `AutoLRS` achieves competitive speed even though it invests a significantly lower search cost that is comparable to the overall model update time associated with the identified LR schedule.

**Transformer:** Transformer (Vaswani et al., 2017) is a neural machine translation (NMT) model that is built upon a multi-head self-attention mechanism to capture the contextual dependencies and achieves promising translation performance. We train Transformer[6] on a standard benchmark, i.e., WMT 2014 English-German dataset, using 8 NVIDIA Tesla V100 GPUs. Following (Vaswani et al., 2017), we use Adam (Kingma & Ba, 2015) with $\beta_1 = 0.9$, $\beta_2 = 0.98$, and $\epsilon = 10^{-9}$. The LR schedule in the original paper starts from a linear warmup of 4,000 steps from 0 to $7e^{-4}$, followed by 96,000 steps of decaying the LR proportionally to $1/\sqrt{t}$ for step-$t$. In `AutoLRS`, we also use the same linear warmup. The current `AutoLRS` does not search LR for warmup steps since warmup

---

[5]All the top-1 accuracy on ImageNet/CIFAR-10/CIFAR-100 we report in the paper is evaluated on all the validation/test data excluding every sample used during training. We report top-1 validation accuracy on ImageNet and top-1 test accuracy on CIFAR-10/CIFAR-100 becasue Imagenet does not disclose the labels of its test set, and CIFAR-10/CIFAR-100 datasets only have the test sets (10,000 test images).

[6]We used the code at https://github.com/tensorflow/models/tree/fd3314e/official/transformer, which achieves 27.3 BLEU (uncased) using the original LR schedule.

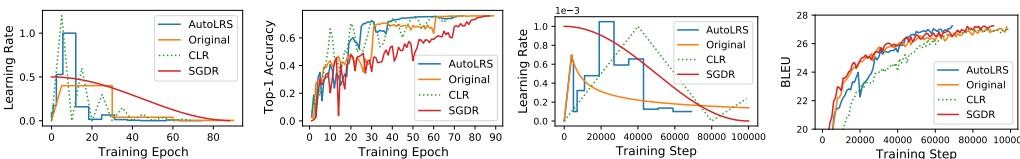

(a) LR on ResNet-50.    (b) Val.Acc. on ResNet-50.    (c) LR for Transformer.    (d) BLEU of Transformer.

Figure 1: Comparison of different LR schedules in training ResNet-50 on ImageNet (a, b), and the Transformer base model (c, d). When training ResNet-50, `AutoLRS`, CLR, SGDR, and the original LR achieve 75.9% top-1 accuracy at epoch 74, 70, 88, and 90, respectively. When training Transformer base, `AutoLRS`, SGDR, and original achieve 27.3 BLEU score (uncased) at step 69,000, 91,000, 98,000, respectively. CLR (the best we were able to find) achieves 27.2 BLEU score at step 99,000.

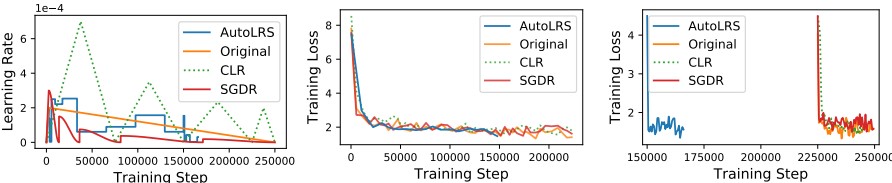

(a) LR schedules (Phase 1 + 2).    (b) Training loss in Phase 1.    (c) Training loss in Phase 2.

Figure 2: Comparison of different LR schedules and training loss in pre-training BERT$_{\text{BASE}}$.

does not have an explicit optimization objective, such as minimizing the validation loss. Warmup usually takes very few steps, and its main purpose is to prevent deeper layers in a DNN from creating training instability (Gotmare et al., 2019). Figure 1c visualizes different LR schedules in training the Transformer model. Their BLEU scores on the test set during training are reported in Figure 1d. Overall, the LR schedule searched by `AutoLRS` yields a $1.32 - 1.43\times$ speedup over the hand-tuned LR schedules. `AutoLRS` consistently achieves a similar amount of speedup over three trials – they achieve 27.3 BLEU score (uncased) at step 69,000, 69,000, and 70,000, respectively. Interestingly, if we continue the LR search of `AutoLRS`, we can get 27.4 BLEU score (uncased) at step 99,000.

**BERT Pre-training:** BERT (Devlin et al., 2019) is a recent model that achieved state-of-the-art results on 11 NLP tasks. It first pre-trains a language representation model on a large text corpus by unsupervised learning and then fine-tunes it for downstream NLP tasks. The BERT$_{\text{BASE}}$ model has 110M parameters, which makes the pre-training phase expensive, and hand-tuning the LR schedule might be impractical. We pre-train BERT$_{\text{BASE}}$ with mixed precision (Micikevicius et al., 2018) on the English Wikipedia and the BooksCorpus dataset[7] (Zhu et al., 2015a). Following the original paper, we use Adam with L2 weight decay of 0.01 and $\beta_1 = 0.9$, $\beta_2 = 0.999$. The pre-training is divided into two phases: Phase 1 includes 90% of the total training steps and uses a sequence length of 128, while Phase 2 uses a sequence length of 512 for the rest 10% of training steps. We apply this two-phase training in the experiments of all LR schedules. We pre-train BERT$_{\text{BASE}}$ on 32 NVIDIA Tesla V100 GPUs using a mini-batch size of 1024 sequences, which is $4\times$ the batch size in the original paper. To adapt the original LR schedule to our batch size, we tried both the linear scaling rule (Goyal et al., 2017) and the square root scaling rule (Krizhevsky, 2014), and found that the square root scaling rule works better while the linear scaling rule made the loss diverge.

As shown in Figure 2, Phase 1 contains 150,000/225,000 steps and Phase 2 contains 16,000/25,000 steps respectively for `AutoLRS` and all baselines, since `AutoLRS` requires much less total steps. In both `AutoLRS` and SGDR, we apply a linear warmup in the first 2,500 steps to make the deeper layers of BERT stable. in Figures 2b and 2c, we report the training loss achieved by different schemes.

We fine-tune the pre-trained models on four downstream NLP tasks: Microsoft Research Paraphrase Corpus (MRPC) for identifying semantic textual similarity (Dolan & Brockett, 2005); Multi-Genre Natural Language Inference (MNLI) for entailment classification (Williams et al., 2018); Corpus of Linguistic Acceptability (CoLA) for predicting whether an English sentence is linguistically acceptable (Warstadt et al., 2019); and Stanford Question Answering Dataset (SQuAD) v1.1 (Rajpurkar et al., 2016). Table 1 reports the after-fine-tuning performance on the four tasks. Since fine-tuning performance is unstable on small datasets like MRPC, we fine-tuned on each task several times and report the best Dev-set performance. It shows that the model pre-trained by `AutoLRS` outperforms

---

[7]We collected books from smashwords.com and built our own dataset with 980M words because the authors of BooksCorpus and no longer have the dataset available for public download (Zhu et al., 2015b).

Table 1: Fine-tuning BERT$_{\text{BASE}}$ that is pre-trained using different LR schedules on 4 downstream tasks. We report the accuracy on the Dev set of MRPC, MNLI, and CoLA, and F1 scores on the Dev set of SQuAD v1.1.

| LR schedule (Phase 1/Phase 2) | MRPC | MNLI | CoLA | SQuAD v1.1 |
|---|---|---|---|---|
| Original (225,000/25,000) | 86.5 | 82.2 | **47.8** | 87.0 |
| CLR (225,000/25,000) | 86.0 | 80.7 | 44.4 | 86.5 |
| SGDR (225,000/25,000) | 84.8 | 81.6 | 38.7 | 86.2 |
| AutoLRS (150,000/16,000) | **88.0** | **82.5** | 47.6 | **87.1** |

Table 2: Performance comparison with LR schedules searched by prior solutions on CIFAR-10 training with VGG-16 (batch size = 128). Note that the hand-tuned LR schedule can achieve 93.70% top-1 test accuracy in 350 epochs. The Runtime column shows how long each method takes on one NVIDIA Titan RTX GPU to find the LR schedule shown in the previous column. The runtime of HD and MARTHE include trying the guideline values of their hyperparameters to get a decent LR schedule.

| Method | Best top-1 accuracy achieved in 350 epochs | Runtime (seconds) |
|---|---|---|
| HD | 91.31% | 187,110 |
| MARTHE | 92.99% | 67,578 |
| Hyperband | 93.24% | 109,454 |
| AutoLRS | **94.13%** | 6,538 |

those using other LR schedules in most downstream tasks and meanwhile achieves a speedup of $1.5\times$. Note AutoLRS consistently achieves this speedup over 3 trials (details in §A.5.5). We also tried pre-training using other LR schedules for fewer steps but the fine-tuning performances were worse. Notably, when we use CLR and SGDR for pre-training BERT$_{\text{BASE}}$, the training loss diverged after 100,000 steps in several trials, even as we decreased the maximal LR and increased the number of steps per cycle. This illustrates how difficult and computationally intensive it is to hand-tune the hyperparameters of existing LR schedules on complicated models and tasks. In contrast, AutoLRS significantly simplifies the process and saves human effort.

**Experimental comparison to prior methods:** Hypergradient descent (HD) (Baydin et al., 2018) is a hypergradient based method to adjust the learning rate in an online fashion by deriving the derivative of the training loss with respect to the learning rate, and performing gradient descent on the learning rate during training. MARTHE (Donini et al., 2020) is a generalization of two hypergradient based methods, HD and RTHO (Franceschi et al., 2017). One distinction between MARTHE and HD is that MARTHE computes the gradient of the validation loss instead of training loss with respect to the learning rate. Hyperband is a multi-armed bandit approach for DNN hyperparameter optimization. We use HD, MARTHE, and Hyperband to tune the LR schedules for CIFAR-10 training with VGG-16, and compare their performance with AutoLRS in Table 2. AutoLRS achieves higher best top-1 test accuracy than the other methods as well as the hand-tuned LR schedule, with much less overhead. Detailed descriptions of these methods and the experimental results are in §A.5.1 and §A.5.3.

## 6 CONCLUSION

We propose an automatic learning-rate schedule method, AutoLRS, as a more efficient and versatile alternative to hand-tuning that can be broadly applied to train different DNNs for tasks in diverse application domains. We break down the sequence optimization to learning rate search for minimizing validation loss in each training stage and then solve this sub-problem by Bayesian optimization (BO). To reduce the cost of BO exploration, we train a light-weight loss-forecasting model from the early-stage training dynamics of BO exploration. AutoLRS achieves a speedup of $1.22\times$, $1.43\times$, and $1.5\times$ on training ResNet-50, Transformer, and BERT compared to their highly hand-tuned schedules.

ACKNOWLEDGMENTS

We would like to thank the anonymous ICLR reviewers for their valuable feedback. We would also like to thank Damien Fay for his suggestions on time series analysis. This work was partially supported by DARPA. For computer time, this research used the resources at ByteDance and the Supercomputing Laboratory at KAUST.

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

# A  APPENDIX

## A.1  BAYESIAN OPTIMIZATION (MORE DETAILS)

BO (Shahriari et al., 2016) is one of the state-of-the-art techniques for black-box optimization. It applies exploration and exploitation to the black-box objective by sequentially and actively querying the function values of some input instances. Specifically, BO uses Gaussian process as a *surrogate model* to fit the black-box objective function $f(\eta)$. It updates a posterior distribution of $f(\eta)$ by using its likelihood on newly evaluated $(\eta, y = f(\eta) + \epsilon)$ pairs, where $y$ is a noisy observation of $f(\eta)$ and is the validation loss after $\tau$ steps in our case. Then, it determines the next LR $\eta$ to evaluate as the one maximizing an *acquisition function*, which is computed from the updated posterior. The *acquisition function* performs a trade-off between exploration and exploitation in evaluating the candidates of LR. BO repeats the above process until achieving a precise posterior predictive distribution of $f(\eta)$.

**Surrogate model (prior):** We apply a commonly used surrogate model — Gaussian process (GP) (Rasmussen & Williams, 2006) as the prior of the black-box objective function in Eq. (2). A GP prior is specified by its mean function $\mu(\cdot)$ and its covariance function (i.e., kernel function) $K(\cdot, \cdot)$. We adopt a common choice $\mu(\cdot) = 0$ and set $K(\cdot, \cdot)$ to be the Matern kernel (Genton, 2001) with smoothness factor $\nu = 2.5$ and length scale $l = 1$, which is defined as

$$K(\eta_i, \eta_j) = \frac{1}{\Gamma(\nu)2^{\nu-1}} \left( \frac{\sqrt{2\nu}\|\eta_i - \eta_j\|_2}{l} \right)^{\nu} K_{\nu} \left( \frac{\sqrt{2\nu}\|\eta_i - \eta_j\|_2}{l} \right), \tag{4}$$

where $K_{\nu}(\cdot)$ is a modified Bessel function and $\Gamma(\cdot)$ is the gamma function, and $K(\eta_i, \eta_j)$ performs a convolution of the unit ball. Comparing to the radial basis function (RBF) kernel which always generates infinitely differentiable functions that might be overly smooth, GP with Matern kernel can control the smoothness of generated functions to be $\lceil \nu \rceil - 1$ times differentiable (Rasmussen & Williams, 2006). This helps to capture the less-smooth local changes. In our case, $\nu = 2.5$ leads to twice-differentiable functions.

**Posterior prediction:** In the following, we will use simplified notations $\eta'_{1:k}$ and $f(\eta'_{1:k})$ for vectors composed of $\{\eta'_i\}_{i=1}^k$ and $\{f(\eta'_i)\}_{i=1}^k$, respectively. The GP prior indicates a Gaussian distribution over function values, i.e., $f(\eta'_{1:k})| \eta'_{1:k} \sim \mathcal{N}(\mathbf{0}, \mathbf{K})$ where $\mathbf{K}_{i,j} = K(\eta'_i, \eta'_j), \forall i, j \in [k]$. After $\tau$ training steps using LR $\eta'_i$, we evaluate the validation loss denoted by $y_i$ as a noisy observation of $f(\eta'_i)$. i.e., $y_i = f(\eta'_i) + \epsilon$ where Gaussian white noise $\epsilon \sim \mathcal{N}(0, \sigma^2)$. Given the noisy observations $y_{1:k}$, we can update the GP posterior of the black-box function $f(\cdot)$ as

$$f(\eta'_{1:k})| \eta'_{1:k}, y_{1:k} \sim \mathcal{N}(y_{1:k}, \mathbf{K} + \sigma^2\mathbf{I}). \tag{5}$$

Given a new LR $\eta$, we can now use the above GP posterior to predict the distribution of $f(\eta)$ by the following reasoning based on Bayes' theorem, i.e.,

$$P(f(\eta)| \eta'_{1:k}, y_{1:k}) = \int P(f(\eta)| f(\eta'_{1:k})) P(f(\eta'_{1:k})| \eta'_{1:k}, y_{1:k}) df(\eta'_{1:k}), \tag{6}$$

which yields the posterior predictive distribution of $f(\eta)$ as

$$\begin{aligned} f(\eta)| \eta'_{1:k}, y_{1:k} \sim \quad & \mathcal{N}(\mu_n(\eta), \sigma_n^2(\eta)), \\ & \mu_n(\eta) \triangleq \mathbf{k}(\mathbf{K} + \sigma^2\mathbf{I})^{-1} y_{1:k}, \\ & \sigma_n^2(\eta) \triangleq K(\eta, \eta) - \mathbf{k}^T(\mathbf{K} + \sigma^2\mathbf{I})^{-1}\mathbf{k}. \end{aligned} \tag{7}$$

where $\mathbf{k}_i = K(\eta, \eta'_i)$. The above result about single LR $\eta$ can be trivially extended to multiple LRs.

**Acquisition function:** Given the posterior predictive distribution of $f(\eta)$ in Eq (7), BO finds the next $\eta'_{i+1}$ to evaluate based on an acquisition function $u_i(\eta)$ defined by the posterior mean $\mu_i(\eta)$ and standard deviation $\sigma_i(\eta)$. A promising acquisition function should balance the trade-off between exploration (i.e., large $\sigma_i(\eta)$) and exploitation (i.e., small $\mu_i(\eta)$). In `AutoLRS`, we use Lower Confidence Bound (LCB) (Cox & John, 1992; Auer, 2002) as our acquisition function. In particular, given $\eta'_{1:k}$ and their corresponding validation loss $y_{1:k}$, we determine the next LR $\eta'_{i+1}$ by minimizing LCB, i.e.,

$$\eta'_{i+1} = \arg\min_{\eta} u_i(\eta), \quad u_i(\eta) \triangleq \mu_i(\eta) - \kappa\sigma_i(\eta), \tag{8}$$

where $\mu_i(\eta)$ and $\sigma_i(\eta)$ were defined in Eq. (7), $\kappa$ is a positive hyper-parameter to balance the trade-off between exploration and exploitation. In experiments, we set $\kappa = 1000$ and it works consistently well.

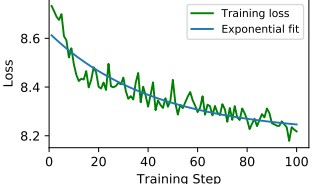
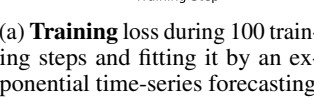
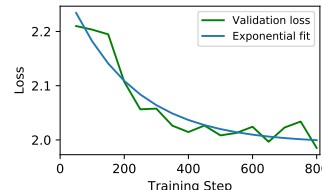
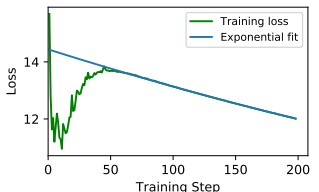

(a) **Training** loss during 100 training steps and fitting it by an exponential time-series forecasting model.

(b) **Validation** loss during 800 training steps and fitting it by an exponential time-series forecasting model.

(c) **A corner case** when exponential model cannot fully capture the non-monotone change of the loss during the first 50 steps.

Figure 3: Fitting the time-series of loss by exponential model when training ResNet-50 on ImageNet.

## A.2 Exponential Model (More Details)

We take two steps to estimate $a, b$, and $c$ in fitting the exponential model $L(t) = a\exp(bt) + c$, based on the validation loss observed in the first $\tau'$ steps, which is represented by $y_t, t = 1, \cdots, \tau'$. First, we reduce the fitting problem to an optimization problem. Define function $g(b)$ as the least squared error between predictions and observations w.r.t. $a$ and $c$. We can write the original fitting problem in the following two-stage form.

$$\min_{b<0} g(b), \ \ g(b) \triangleq \min_{a,c} \sum_{t=1}^{\tau'} \left( a\exp(bt) + c - y_t \right)^2 \tag{9}$$

It is a 1-dimensional optimization problem. Moreover, with $b$ fixed, the minimization problem w.r.t. $a, c$ is a linear regression problem that has a closed-form solution. Hence, we apply a simple gradient descent method that starts from an initial $b$, computes the linear least squares w.r.t. $a, c$ under $b$, search for the next $b$ by the gradient descent method, and repeats these two steps. Thereby, in practice we can achieve a fast decrease on the regression error. In addition, to enforce the negative constraint for $b$, we re-parameterize it to be $b \leftarrow -\exp(b')$. The problem now reduces to

$$\min_{b'} \min_{a,c} \sum_{t=1}^{\tau'} \left( a\exp(-\exp(b')t) + c - y_t \right)^2 \tag{10}$$

Although there might exist other possible strategies to optimize Eq. (9), we find the above method is stable and fast in reducing the regression error and thus keeps the fitting process highly efficient.

We empirically test whether the exponential model obtained by our method can ideally fit the loss time-series in different cases. Figure 3a and Figure 3b are two typical examples fitting the time series of training loss and validation loss by the proposed model. They show that the model can precisely predict the main trends of the time-varying loss, though ruling out some less informative noises.

In Figure 3c, we also show a rare corner case when the model fails to fit the increasing loss in early steps. However, the loss-increasing stage usually does not last long and thus the inaccuracy is not so harmful to the prediction of later-stage loss, which is our major goal since $\tau$ is usually larger than the length of loss-increasing stage. To overcome such corner cases and outliers in the observed validation loss series, we present a pre-processing strategy to make stable exponential fitting in §A.3. Every time we predict the validation loss after $\tau$ steps, we first pre-process the loss observed in the $\tau'$ steps, and then fit the pre-processed loss series with the exponential model.

In our empirical study, we also tried other more sophisticated time-series forecasting models including Holt-Winters, autoregressive integrated moving average (ARIMA) and singular spectrum analysis (SSA). We show two examples to compare their performance with our simple exponential prediction model in Figure 4. Some prior works also fit and predict learning curves (Swersky et al., 2014; Domhan et al., 2015; Klein et al., 2017; Dai et al., 2019) for early termination of evaluations of poorly-performing hyperparameters when doing DNN hyperparameter optimization, but they need non-negligible time for training their models and performing inference. They are much more computationally intensive than our lightweight exponential prediction model, and this makes them less practical to be used in automatic LR schedule tuning.

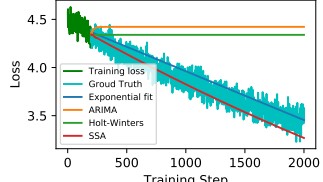 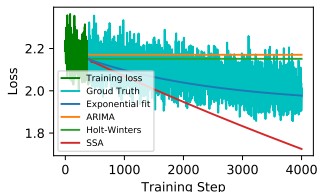

(a) Predict the training loss after 2000 steps.      (b) Predict the training loss after 4000 steps.

Figure 4: Examples of forecasting the loss series by various time-series forecasting models when training ResNet-50 on ImageNet. Our simple exponential prediction model yields the least mean squared error (MSE) among all the models.

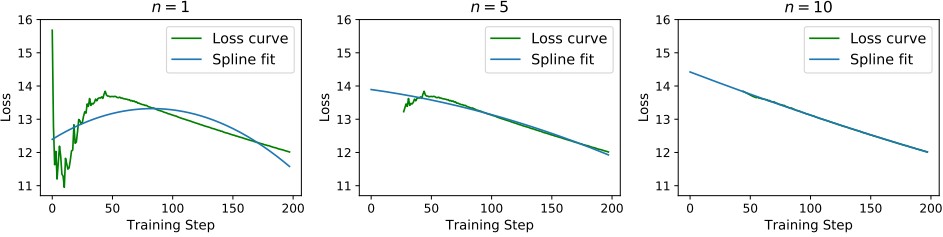

Figure 5: The loss sequence in Figure 3c and its quadratic spline smoothing result after 1, 5, and 10 iterations of our spline smoothing.

### A.3    PRE-PROCESS LOSS SERIES BY ITERATIVE SPLINE SMOOTHING

We show a corner case in Figure 3c where the loss decreases rapidly at first, then increases for a while, but finally decreases stably. It might be a result of a large LR or happens when escaping from a possibly poor local minimum or a saddle point (Goodfellow et al., 2016). Our exponential model cannot fully capture the loss change in early steps of this case. But we also consistently observe in our experiments that the early instability of loss only lasts for at most hundreds of steps after we switch to a new LR.

Nevertheless, we find that adding a pre-processing step to eliminate the noises, anomalies, and corner cases in the observed validation loss series makes the exponential fitting easier and more stable. Hence, we propose to apply an iterative spline smoothing to the validation loss observed in $\tau'$ steps before training the forecasting model. In particular, when evaluating a LR $\eta$ for a training stage, we firstly run $\tau'$ training steps and fit the observed sequence of validation loss by a quadratic spline. We do such spline smoothing for multiple iterations. At the end of each iteration, we remove the loss values that are among the farthest 3% from the spline smoothing results if they are collected in the first $\tau'/2$ steps (when the corner cases like the one in Figure 3c might happen). So the next iteration's spline smoothing only aims to fit the rest loss values. After certain number of iterations, we use the final spline smoothing values to train the exponential forecasting model.

Empirically, we find that 10 iterations of the above spline smoothing are necessary before training the exponential forecasting model. Figure 5 shows the loss sequence from Figure 3c before smoothing and after 1, 5 and 10 iterations of smoothing. As shown in the plots, the iterative spline smoothing can effectively remove the unnecessary noise and unstable changes during the early phase.

### A.4    POSTERIOR LEARNED BY BO

Figure 6 shows how the BO posterior gradually learns an increasingly more accurate estimation of the underlying black-box objective. We also visualize the "learning progress" of BO in an earlier stage and a later stage during training. It shows that in both early and late stages, by exploring more LRs, BO can achieve a more accurate posterior estimation of the objective function, and $k = 10$ suffices to obtain a satisfying estimate. Moreover, the posteriors in the later stages have much smaller variance/uncertainty than in the earlier stages.

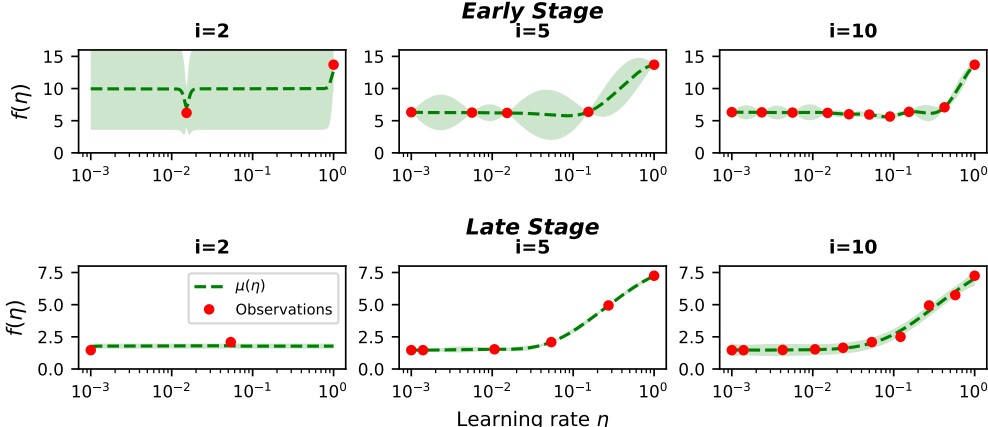

Figure 6: BO's posterior of the black-box objective function after exploring $i$ LRs (red dots ●) determined by Eq. (3) at an *early stage* and a *late stage* during the training of ResNet-50 on ImageNet. The dashed lines show the mean function $\mu_i(\eta)$ (indicating the predicted validation loss of applying LR $\eta$ for $\tau$ steps) and the shaded areas show the standard deviation $\sigma_i(\eta)$ (indicating the prediction uncertainty) in the form of $\mu_i(\eta) \pm \sigma_i(\eta)$.

## A.5 EXPERIMENTS (MORE DETAILS)

### A.5.1 ANALYSIS OF ONLINE LEARNING RATE ADAPTATION WITH HYPERGRADIENT-BASED METHODS

Hypergradient descent (HD) (Baydin et al., 2018) is a method to adjust the learning rate in an online fashion by performing gradient descent on the learning rate at the same time as the underlying DNN is optimized. For simplicity, we rewrite Eq. (1), which performs mini-batch SGD updates on model weights $\theta$ at each step $t$, as:

$$\theta_{t+1} = \theta_t - \eta_t \nabla L(\theta_t), \tag{11}$$

where $\eta_t$ is the learning rate (LR) at step $t$ and $\nabla L(\theta_t)$ denotes the gradient of the loss function $L$ w.r.t. the model weights $\theta_t$ at step $t$. By making the assumption that the optimal value of LR does not change much between two consecutive iterations, HD derives the partial derivative of the loss function $L$ with respect to the learning rate $\eta$:

$$\frac{\partial L(\theta_t)}{\partial \eta} = \nabla L(\theta_t) \frac{\partial(\theta_{t-1} - \eta \nabla L(\theta_{t-1}))}{\partial \eta} = \nabla L(\theta_t)(-\nabla L(\theta_{t-1})) \tag{12}$$

An update rule for the learning rate is constructed as:

$$\eta_{t+1} = \eta_t - \beta \frac{\partial L(\theta_t)}{\partial \eta} = \eta_t + \beta \nabla L(\theta_t) \nabla L(\theta_{t-1}), \tag{13}$$

which introduces a hyperparameter $\beta$ as the hypergradient learning rate. Updating the learning rate is a single vector multiplication between the gradient of the model weights at the previous step and the one at the current step. By updating both the learning rate $\eta_t$ and the model weights $\theta_t$ using Eq.(13) and Eq.(11) in each step, the HD algorithm performs gradient descent on both learning rate and the model weights during training.

HD can be applied to optimizers including SGD, SGD with Nesterov momentum, and Adam. The original paper empirically shows that these optimizers equipped with HD are much less sensitive to the choice of the initial regular learning rate and the convergence rate is improved on a set of tasks. However, the paper only compares HD with constant LR baselines on small models and small datasets. To study how HD compares to hand-tuned LR schedules on larger models and datasets, we train VGG-16 (Simonyan & Zisserman, 2015) and ResNet-50 neural networks on the CIFAR-10 image recognition dataset (Krizhevsky & Hinton, 2009) with a mini-batch size of 128 using a PyTorch implementation.[8] A hand-tuned LR schedule consists of a total of 350 epochs, starting with 0.1 and multiplying the learning rate by 0.1 at epoch 150 and 250. This hand-tuned LR schedule can achieve around 93.70% and 95.56% top-1 accuracy on the test set for VGG-16 and ResNet-50, respectively when we train the models on one NVIDIA Titan RTX GPU. We apply SGD with HD (SGD-HD)[9] to train the two models, sweep all the guideline values of the two hyperparameters (regular LR and

---

[8]https://github.com/kuangliu/pytorch-cifar
[9]https://github.com/gbaydin/hypergradient-descent

hypergradient LR) in SGD-HD, and report the best top-1 accuracy that SGD-HD can achieve for VGG-16 and ResNet-50 within 500 epochs in Table 4 and Table 5. We have three observations: (1) Hypergradient descent is very sensitive to the selection of the regular LR and the hypergradient LR. The top-1 accuracy ranges from 10.00% to 91.80% for VGG-16 and ranges from 10.00% to 92.52% for ResNet-50, with all suggested values of the two hyperparameters. (2) It cannot match the top-1 accuracy achieved with hand-tuned LR schedules: the best top-1 accuracy it can achieve among all the different hyperparameter settings are 1.90% and 3.04% behind the accuracy achieved with hand-tuned LR schedules for VGG-16 and ResNet-50, even though we ran each of them 150 epochs more than the hand-tuned LR schedule. (3) It is prone to overfitting. For example, when using regular $LR = 10^{-3}$ and hypergradient $LR = 10^{-5}$ to train VGG-16, the top-1 accuracy is only $90.74\%$ while the training accuracy is already $99.98\%$.

MARTHE (Donini et al., 2020) adaptively interpolates between two hypergradient based methods, HD and RTHO (Franceschi et al., 2017), and it computes the gradient of the loss function on the validation set instead of training set w.r.t. the learning rate. Besides the two hyperparameters in HD, MARTHE introduces another hyperparameter $\mu$ that controls how quickly past history is forgotten. We sample $\mu$ between 0.9 and 0.999, sample the hypergradient LR in $[10^{-3}, 10^{-6}]$ log-uniformly, and set the initial LR to 0.1, as how the MARTHE paper set its hyperparameters for training VGG-11 on CIFAR-10. We apply SGD with MARTHE[10] to train VGG-16 on CIFAR-10. The best top-1 accuracy MARTHE can achieve among all the hyperparameter settings in 350 epochs is 92.99%, which is 0.71% lower than the accuracy achieved with the hand-tuned LR schedule.

### A.5.2 SENSITIVITY TEST OF $\tau_{\max}$ IN AutoLRS AND MEASURE OF VARIABILITY

Recall from §4.4 that `AutoLRS` starts with $\tau = 1000$ and $\tau' = 100$, and doubles them after every stage until it reaches $\tau_{\max}$. We test the sensitivity of `AutoLRS` to this hyperparameter, $\tau_{\max}$, by comparing the generated LR schedules with different $\tau_{\max}$ values for the VGG-16 neural network on CIFAR-10 as in §A.5.1. The LR search interval $(\eta_{\min}, \eta_{\max})$ we use is $(10^{-3}, 10^{-1})$. We report the training epochs to reach the target $93.70\%$ top-1 accuracy using the LR schedules generated among 5 trials for different $\tau_{\max}$ values in Table 6. `AutoLRS` with different $\tau_{\max}$ values can consistently achieve the target top-1 accuracy achieved with the hand-tuned LR schedule (i.e., $93.70\%$) in fewer training steps. We also see that the best `AutoLRS`-generated LR schedule can achieve $94.13\%$ top-1 accuracy within 350 training epochs (excluding the costs of the LR search).

In the last column of Table 6, we report the mean and standard deviation of the top-1 accuracy achieved by AutoLRS over 5 trials for each $\tau_{\max}$. To further measure the variability of `AutoLRS`, we train VGG-16 on CIFAR-100 (Krizhevsky et al.) with a mini-batch size of 128. A carefully hand-tuned LR schedule consists of a total of 200 epochs, starting with 0.1 and dividing the learning rate by 5 at epoch 60, 120, and 160.[11] This hand-tuned LR schedule can achieve 72.93% top-1 accuracy. We train VGG-16 on CIFAR-100 for 200 epochs with `AutoLRS` for 10 trials using different random seeds, and report the top-1 accuracy they achieve in Table 9. The LR search interval $(\eta_{\min}, \eta_{\max})$ we use is $(10^{-3}, 10^{-1})$, and $\tau_{\max}$ is set to 8000. The top-1 accuracy achieved by `AutoLRS`-generated LR schedules over 10 trials are distributed with a mean of 73.05% and a standard deviation of 0.14%. The best `AutoLRS`-generated LR schedule can achieve 73.30% top-1 accuracy, which is 0.37% higher than the accuracy achieved using the hand-tuned LR schedule.

### A.5.3 LEARNING RATE SCHEDULE SEARCH WITH HYPERBAND

Hyperband is a multi-armed bandit approach for DNN hyperparameter optimization. It dynamically allocates resources to randomly sampled configurations and uses successive halving (Jamieson & Talwalkar, 2016) to early stop poorly-performing configurations. We attempt to use Hyperband to optimize the LR schedule on CIFAR-10 training with VGG-16 by searching for an exponential decay LR schedule, which can be parameterized with an initial learning rate and a decay factor. The learning rate is decayed by the decay factor every epoch. Exponential decay is a commonly used LR schedule and is also used in other DNN hyperparameter optimization methods (Falkner et al., 2018). We use the search space of $(10^{-3}, 10^{-1})$ for the initial LR, and the search space of $(0.9, 1)$ for the decay rate. The decay rate is uniformly random sampled, and the initial LR is uniformly random sampled in its log-scale space. We use the default setting of Hyperband that sets the maximum epochs that can be

---

[10]https://github.com/awslabs/adatune
[11]https://github.com/weiaicunzai/pytorch-cifar100

allocated to a single configuration to 350 and discards two-thirds of the configurations in each round of successive halving. This results in evaluating 384 configurations with different numbers of epochs with a total of 12600 epochs, which has a computational overhead of $36\times$ compared to a single run of training with the hand-tuned LR schedule. The best configuration found by Hyperband achieves 93.24% top-1 accuracy, which is 0.46% lower than the accuracy achieved with the hand-tuned LR schedule.

### A.5.4 ABLATION STUDY

To illustrate the effects of the exponential model and BO of `AutoLRS`, we perform ablation studies using the VGG-16 neural network on CIFAR-10 as in §A.5.1.

**Exponential model:** What if we remove the exponential forecasting model and simply use the validation loss at $\tau'$ step to update the BO posterior? Will the LR schedules generated by `AutoLRS` be significantly worse? We apply `AutoLRS` without the exponential forecasting to find the LR schedules for VGG-16 on CIFAR-10. With $\tau_{\max}$ being chosen from the set of $\{4000, 8000, 16000\}$, the best top-1 test accuracy that `AutoLRS` can achieve within 350 training epochs are 91.73%, 92.59%, 92.24%, respectively. Therefore, it is unable to match the target accuracy in reasonable training steps without the exponential forecasting model. The reason for this is that the objective of BO has become to minimize the validation loss at the $\tau'$ step, which will lead to short-horizon issues (Wu et al., 2018). As a consequence, it tends to select a conservative LR, which is often a small LR around $\eta_{\min}$ in the late stages. In contract, with the exponential forecasting model, the goal of the BO is to find the LR that minimizes the predicted validation loss in $\tau$ steps. This allows the LR selected in the current stage to be higher than that in the past stages, and the loss to even increase in a short period of time, as long as the predicted loss in $\tau$ steps is low. This phenomenon can be seen in Figure 1 and Figure 2.

**Bayesian Optimization:** What if we replace BO in `AutoLRS` with random search or grid search? Will the LR schedules generated by `AutoLRS` get worse? We replace the BO part in `AutoLRS` with random search and grid search while keeping the exponential forecasting part of it, and apply it to find the LR schedules for VGG-16 on CIFAR-10. The LR search interval is $(10^{-3}, 10^{-1})$, the same as in §A.5.2. Table 7 and Table 8 show the results of random search and grid search with different $\tau_{\max}$ values, respectively. We observe that both random search and grid search have at least one trial that fails to match the hand-tuned LR schedule to achieve 93.70% top-1 test accuracy within 350 epochs (denoted by N/A in the tables). The top-1 accuracy achieved on average across trials in 350 epochs by random search and grid search is 0.09% and 0.24% behind `AutoLRS` with BO, respectively. We also replace BO with grid search and apply it to find the LR schedules for VGG-16 on CIFAR-100. The top-1 accuracy achieved over 10 trials are distributed with a mean of 72.63% and a standard deviation of 0.56%. Compared to the `AutoLRS`-generated LR schedules in Table 9, the mean of the grid search accuracy is out of two standard deviations from the BO accuracy 73.05%±0.14%.

### A.5.5 `AutoLRS` FINE-TUNING RESULTS OF BERT$_{\text{BASE}}$ ACROSS 3 TRIALS

We pre-trained BERT$_{\text{BASE}}$ with `AutoLRS` for 3 trials, and report their fine-tuning results in Table 3.

Table 3: Fine-tuning results of BERT$_{\text{BASE}}$ models pre-trained with `AutoLRS` for 3 trials. Accuracy scores on the Dev set are reported for MRPC, MNLI, and CoLA. F1 scores on the Dev set are reported for SQuAD v1.1.

|         | MRPC | MNLI | CoLA | SQuAD v1.1 |
|---------|------|------|------|------------|
| Trial 1 | 88.0 | 82.5 | 47.6 | 87.1       |
| Trial 2 | 88.0 | 82.7 | 46.5 | 87.0       |
| Trial 3 | 87.8 | 82.3 | 47.0 | 86.6       |

Table 4: The accuracy information of tuning the regular LR and the hypergradient LR of SGD-HD for CIFAR-10 training with VGG-16 (batch size = 128). We train the model for 500 epochs using SGD-HD with each suggested value for the regular LR and the hypergradient LR, and report the best top-1 test accuracy it can achieve, its corresponding training accuracy, and the epoch number. Note that a hand-tuned LR schedule can achieve 93.70% top-1 test accuracy in 350 epochs.

| regular LR | hypergradient LR | Top-1 Test Accuracy | Training Accuracy | Epoch |
|---|---|---|---|---|
| $10^{-6}$ | $10^{-6}$ | 86.13% | 99.77% | 438 |
| $10^{-6}$ | $10^{-5}$ | 88.79% | 99.98% | 480 |
| $10^{-6}$ | $10^{-4}$ | 86.31% | 98.10% | 494 |
| $10^{-6}$ | $10^{-3}$ | 90.70% | 99.95% | 499 |
| $10^{-6}$ | $10^{-2}$ | 10.30% | 9.90% | 40 |
| $10^{-6}$ | $10^{-1}$ | 10.00% | 10.00% | 1 |
| $10^{-5}$ | $10^{-6}$ | 86.14% | 99.73% | 394 |
| $10^{-5}$ | $10^{-5}$ | 88.49% | 99.95% | 448 |
| $10^{-5}$ | $10^{-4}$ | 87.67% | 98.78% | 483 |
| $10^{-5}$ | $10^{-3}$ | 88.70% | 99.49% | 469 |
| $10^{-5}$ | $10^{-2}$ | 10.22% | 9.92% | 170 |
| $10^{-5}$ | $10^{-1}$ | 10.00% | 10.00% | 1 |
| $10^{-4}$ | $10^{-6}$ | 86.09% | 99.84% | 481 |
| $10^{-4}$ | $10^{-5}$ | 88.82% | 99.94% | 304 |
| $10^{-4}$ | $10^{-4}$ | 86.63% | 95.37% | 479 |
| $10^{-4}$ | $10^{-3}$ | 10.22% | 10.13% | 1 |
| $10^{-4}$ | $10^{-2}$ | 10.02% | 10.00% | 1 |
| $10^{-4}$ | $10^{-1}$ | 10.00% | 10.00% | 1 |
| $10^{-3}$ | $10^{-6}$ | 86.13% | 99.73% | 406 |
| $10^{-3}$ | $10^{-5}$ | 88.78% | 99.94% | 346 |
| $10^{-3}$ | $10^{-4}$ | 90.74% | 99.98% | 484 |
| $10^{-3}$ | $10^{-3}$ | 44.12% | 43.02% | 500 |
| $10^{-3}$ | $10^{-2}$ | 88.48% | 99.55% | 467 |
| $10^{-3}$ | $10^{-1}$ | 10.00% | 10.00% | 1 |
| $10^{-2}$ | $10^{-6}$ | 91.69% | 99.97% | 389 |
| $10^{-2}$ | $10^{-5}$ | 88.53% | 99.89% | 397 |
| $10^{-2}$ | $10^{-4}$ | 89.11% | 99.92% | 484 |
| $10^{-2}$ | $10^{-3}$ | 10.07% | 9.90% | 265 |
| $10^{-2}$ | $10^{-2}$ | 10.00% | 10.02% | 1 |
| $10^{-2}$ | $10^{-1}$ | 10.00% | 9.99% | 1 |
| $10^{-1}$ | $10^{-6}$ | **91.80%** | **99.93%** | 476 |
| $10^{-1}$ | $10^{-5}$ | 91.48% | 99.85% | 317 |
| $10^{-1}$ | $10^{-4}$ | 88.81% | 99.57% | 499 |
| $10^{-1}$ | $10^{-3}$ | 90.42% | 99.80% | 393 |
| $10^{-1}$ | $10^{-2}$ | 11.24% | 10.45% | 1 |
| $10^{-1}$ | $10^{-1}$ | 10.00% | 10.02% | 1 |

Table 5: The accuracy information of tuning the regular LR and the hypergradient LR of SGD-HD for CIFAR-10 training with ResNet-50 (batch size = 128). We train the model for 500 epochs using SGD-HD with each suggested value for the regular LR and the hypergradient LR, and report the best top-1 test accuracy it can achieve, its corresponding training accuracy, and the epoch number. Note that a hand-tuned LR schedule can achieve $95.56\%$ top-1 test accuracy in 350 epochs.

| regular LR | hypergradient LR | Top-1 Test Accuracy | Training Accuracy | Epoch |
|---|---|---|---|---|
| $10^{-6}$ | $10^{-6}$ | 83.67% | 99.71% | 410 |
| $10^{-6}$ | $10^{-5}$ | 88.75% | 99.44% | 490 |
| $10^{-6}$ | $10^{-4}$ | 83.77% | 99.68% | 494 |
| $10^{-6}$ | $10^{-3}$ | 71.03% | 72.14% | 491 |
| $10^{-6}$ | $10^{-2}$ | 10.11% | 10.03% | 261 |
| $10^{-6}$ | $10^{-1}$ | 10.0% | 10.0% | 1 |
| $10^{-5}$ | $10^{-6}$ | 83.99% | 99.64% | 420 |
| $10^{-5}$ | $10^{-5}$ | 89.15% | 99.97% | 460 |
| $10^{-5}$ | $10^{-4}$ | 10.12% | 9.95% | 206 |
| $10^{-5}$ | $10^{-3}$ | 19.73% | 18.53% | 13 |
| $10^{-5}$ | $10^{-2}$ | 10.03% | 9.98% | 137 |
| $10^{-5}$ | $10^{-1}$ | 10.0% | 10.0% | 1 |
| $10^{-4}$ | $10^{-6}$ | 84.98% | 99.85% | 488 |
| $10^{-4}$ | $10^{-5}$ | 89.27% | 99.94% | 482 |
| $10^{-4}$ | $10^{-4}$ | 84.36% | 97.78% | 424 |
| $10^{-4}$ | $10^{-3}$ | 88.72% | 99.84% | 484 |
| $10^{-4}$ | $10^{-2}$ | 10.00% | 10.00% | 1 |
| $10^{-4}$ | $10^{-1}$ | 10.00% | 10.00% | 1 |
| $10^{-3}$ | $10^{-6}$ | 83.22% | 99.81% | 487 |
| $10^{-3}$ | $10^{-5}$ | 88.56% | 99.98% | 492 |
| $10^{-3}$ | $10^{-4}$ | 86.00% | 97.32% | 440 |
| $10^{-3}$ | $10^{-3}$ | 10.10% | 9.76% | 367 |
| $10^{-3}$ | $10^{-2}$ | 42.80% | 40.11% | 497 |
| $10^{-3}$ | $10^{-1}$ | 10.00% | 10.00% | 1 |
| $10^{-2}$ | $10^{-6}$ | 92.40% | 99.99% | 459 |
| $10^{-2}$ | $10^{-5}$ | 88.51% | 99.98% | 440 |
| $10^{-2}$ | $10^{-4}$ | 90.72% | 99.91% | 452 |
| $10^{-2}$ | $10^{-3}$ | 10.19% | 9.64% | 315 |
| $10^{-2}$ | $10^{-2}$ | 10.05% | 9.99% | 8 |
| $10^{-2}$ | $10^{-1}$ | 10.00% | 10.00% | 1 |
| $10^{-1}$ | $10^{-6}$ | 92.18% | 99.97% | 487 |
| $10^{-1}$ | $10^{-5}$ | **92.52%** | **99.97%** | 494 |
| $10^{-1}$ | $10^{-4}$ | 87.74% | 99.86% | 492 |
| $10^{-1}$ | $10^{-3}$ | 84.32% | 97.23% | 477 |
| $10^{-1}$ | $10^{-2}$ | 10.00% | 10.11% | 1 |
| $10^{-1}$ | $10^{-1}$ | 10.00% | 10.00% | 1 |

Table 6: Performance of `AutoLRS` with different $\tau_{max}$ values for CIFAR-10 training with VGG-16 (batch size = 128). Note that a hand-tuned LR schedule can achieve 93.70% top-1 test accuracy in 350 epochs. We report the top-1 accuracy achieved within 350 epochs for each trial, and the mean and standard deviation of the top-1 accuracy achieved by `AutoLRS` over 5 trials for each $\tau_{max}$.

| $\tau_{max}$ | Trial Number | Epoch to 93.70% Top-1 Accuracy | Top-1 Accuracy Achieved | Mean±std |
|---|---|---|---|---|
| 4000 | Trial 1 | 108 | 94.13% | |
| | Trial 2 | 181 | 93.96% | |
| | Trial 3 | 223 | 94.07% | 94.01%±0.13% |
| | Trial 4 | 315 | 93.82% | |
| | Trial 5 | 287 | 94.09% | |
| 8000 | Trial 1 | 115 | 94.03% | |
| | Trial 2 | 265 | 93.92% | |
| | Trial 3 | 203 | 93.94% | 93.96%±0.07% |
| | Trial 4 | 194 | 94.02% | |
| | Trial 5 | 305 | 93.87% | |
| 16000 | Trial 1 | 229 | 93.77% | |
| | Trial 2 | 250 | 93.95% | |
| | Trial 3 | 267 | 93.73% | 93.80%±0.10% |
| | Trial 4 | 313 | 93.71% | |
| | Trial 5 | 330 | 93.82% | |

Table 7: Experimental results after replacing BO in `AutoLRS` with random search for CIFAR-10 training with VGG-16 (batch size = 128). We also report the top-1 accuracy achieved within 350 epochs for each trial.

| $\tau_{max}$ | Trial Number | Epoch to 93.70% Top-1 Accuracy | Top-1 Accuracy Achieved |
|---|---|---|---|
| 4000 | Trial 1 | 199 | 93.80% |
| | Trial 2 | 209 | 93.97% |
| | Trial 3 | 298 | 93.84% |
| 8000 | Trial 1 | 344 | 93.71% |
| | Trial 2 | 225 | 93.98% |
| | Trial 3 | 175 | 93.91% |
| 16000 | Trial 1 | N/A | 93.64% |
| | Trial 2 | 316 | 93.96% |
| | Trial 3 | 310 | 93.86% |

Table 8: Experimental results after replacing BO in `AutoLRS` with grid search for CIFAR-10 training with VGG-16 (batch size = 128). We also report the top-1 accuracy achieved within 350 epochs for each trial.

| $\tau_{max}$ | Trial Number | Epoch to 93.70% Top-1 Accuracy | Top-1 Accuracy Achieved |
|---|---|---|---|
| 4000 | Trial 1 | 304 | 93.88% |
| | Trial 2 | 233 | 93.88% |
| | Trial 3 | 180 | 93.91% |
| 8000 | Trial 1 | 239 | 93.72% |
| | Trial 2 | 296 | 93.95% |
| | Trial 3 | N/A | 93.32% |
| 16000 | Trial 1 | N/A | 93.02% |
| | Trial 2 | 153 | 93.78% |
| | Trial 3 | 288 | 93.70% |

Table 9: Top-1 test accuracy achieved by `AutoLRS`-generated LR schedules for CIFAR-100 training with VGG-16 over 10 trials.

| | | | | |
|---|---|---|---|---|
| 73.12% | 73.20% | 72.90% | 72.93% | 73.03% |
| 73.16% | 73.30% | 72.85% | 73.00% | 72.97% |

