# OpenReview forum: "AutoLRS: Automatic Learning-Rate Schedule by Bayesian Optimization on the Fly"
_ICLR.cc/2021/Conference — ICLR 2021 Poster_

### Official Review · AnonReviewer4 · 2020-10-26
**Very Interesting Application of Bayesian Optimization**

**Rating:** 7
**Confidence:** 4

**Review:**

Summary:
This paper uses Bayesian optimization (BO) to dynamically tune the learning rate during the course of training of DNNs.  In every stage of training, the algorithm firstly uses BO to explore different learning rates with the help of a parametric exponential model for learning rate extrapolation, and then applies the selected learning rate at the current stage. The algorithm is applied to the training of state-of-the-art DNN models, and is shown to outperform the original learning rate schedules, as well as other methods for learning rate scheduling.

Strong points:
- I think this is a very interesting and practical application of Bayesian optimization (BO).
- The way BO is used for learning rate selection in every stage is very clever, especially the trick of reverting back to the checkpoint model parameter before the evaluation of every LR, which resolves the non-stationarity issue due to evolving model parameters.
- The experiments are conducted using state-of-the-art DNN models such as transformers and BERT.
- The paper is in general vey well written, although some places describing the algorithm could be improved as specified below.

Weak points/questions:
- I think the brief descriptions of the algorithm especially in the Abstract and Introduction are not very accurate and sometimes confusing (especially the 3rd paragraph of the Introduction). I couldn't really tell how the algorithm works until I've read Algorithm 1. I feel it would be easier to understand if these descriptions clearly say that: in every stage, BO is firstly used to select the learning rate, where each learning rate evaluation during BO starts at the same model parameter checkpoint and is early-stopped using an exponential model; after BO, the selected optimal learning rate is used for throughout the current stage.
- Since your algorithm incurs additional computational cost due to BO, I think it would be a more fair comparison if you compare the runtime (instead of training epochs/steps as in Figures 1 and 2) of different algorithms. I wonder whether the proposed algorithm still has an advantage after taking into account the additional computational cost due to BO.
- In the experiments, the number of BO iterations is selected as k=10. I think this is a small number of iterations for BO. Given that BO needs to perform exploration at the initial iterations, I wonder is how your algorithm will perform if you replace BO by pure exploration (i.e., simple random search)?
- The proposed Algorithm 1 seems extremely similar to Refs [1] and [2] given below, and hence they should be referenced. Both [1] and [2] also use an exponentially decaying kernel to model the evolution of the learning curve, and hence early-stop some evaluations of hyperparameters (learning rate).

[1] Bayesian optimization meets Bayesian optimal stopping, ICML 2019.
[2] Freeze-thaw Bayesian optimization, Arxiv, 2014.

---

> ### Author Response · Authors · 2020-11-18
> **Response to Reviewer 4**
>
> We would like to thank you for your kind words about our work. We are glad that you find our practical application of BO to dynamic learning rate tuning very interesting. We want to address your concerns and answer your questions here:
>
> - \> “I think the brief descriptions of the algorithm especially in the Abstract and Introduction are not very accurate and sometimes confusing.”\
> Thank you for pointing this out! We have updated the introduction of the revision according to your suggestion. We hope it makes the brief descriptions of the algorithm clearer.
>
> - \> “Since your algorithm incurs additional computational cost due to BO, I think it would be a more fair comparison if you compare the runtime (instead of training epochs/steps as in Figures 1 and 2) of different algorithms.”\
> We calculated the computational time spent on BO in each stage. In each stage, we only update the GP posterior 10 times. In total, the time spent on BO in each stage is only 38ms, which is negligible compared to the time spent on training the DNN model for $\tau$ steps. For example, when we train VGG-16 on CIFAR-100, a training stage with 1000 training steps takes 50 seconds, which is more than $1000\times$ the BO cost. The runtime overhead of AutoLRS is the training cost associated with the generated schedule due to the BO exploration, plus the cost of the infrequent evaluation of validation loss on a small subset of the validation set during the BO search in the later stages as discussed in footnote 4. In conclusion, the runtime of AutoLRS is slightly more than twice the training cost associated with the generated schedule.\
> In the latest revision of the paper, we additionally report the Runtime in a column of Table 2 in Section 5. This column shows how long each method takes on one NVIDIA Titan RTX GPU to find a good LR schedule for training VGG-16 on CIFAR-10. AutoLRS reduces the runtime by 28.6$\times$, 10.3$\times$, 16.7$\times$ compared to Hypergradient, MARTHE, and Hyperband, respectively. More importantly, the LR schedule found by AutoLRS is of higher quality. The runtime of the Hypergradient and MARTHE is more costly because they are sensitive to the hyperparameters and thus one needs to try many hyperparameter combinations to find a decent LR schedule. The overhead of Hyperband is due to the fact that it needs to evaluate various configurations with different numbers of epochs. Even with successive halving (early stopping), it still yields a large overhead.
>
> - \> “I wonder how your algorithm will perform if you replace BO by pure exploration?”\
> We conducted an additional ablation study to confirm the effectiveness of BO in our algorithm. We have included the results of the ablation study in Appendix A.5.4, Tables 7-8. Compared to the performance of AutoLRS in Table 6, the highest top-1 accuracy achieved in 350 epochs by random search and grid search are 0.09\% and 0.24\% behind AutoLRS with BO respectively on average across trials.
>
> - \> “The proposed Algorithm 1 seems extremely similar to Refs [1] and [2] given below, and hence they should be referenced.”\
> Thank you for the pointers! We have added the missing references to Appendix A.2 in Section 4.
>
> Thank you for your review! Please let us know if you have any concerns or questions.
>
> References\
> [1] Dai, Zhongxiang, Haibin Yu, Bryan Kian Hsiang Low, and Patrick Jaillet. "Bayesian optimization meets Bayesian optimal stopping." ICML, 2019.\
> [2] Swersky, Kevin, Jasper Snoek, and Ryan Prescott Adams. "Freeze-thaw Bayesian optimization." arXiv preprint arXiv:1406.3896, 2014.

---

> > ### Comment · AnonReviewer4 · 2020-11-23
> > **Appreciate Authors' Response**
> >
> > I appreciate the authors' response and additional experiments.
> > My main concerns were the additional computational cost introduced by BO, and whether BO shows an advantage over pure exploration/random search. The authors addressed my concerns very well with additional experiments, since they showed that the additional cost brought by BO is negligible, and using BO indeed brings benefit compared with using random search. So I've decided to raise my score.

---

### Official Review · AnonReviewer3 · 2020-10-27
**The paper is well-written and has an interesting approach. Minor concerns regarding hidden hyperparameters and statistical significance of empirical results.**

**Rating:** 7
**Confidence:** 4

**Review:**


## Overview
---
Training deep neural networks is typically done using gradient-based methods with either pre-defined learning-rate schedules or off-the-shelf adaptive optimizers (such as Adam). The former can not reliably align with the non-linear loss landscape, while the latter add additional hyperparameters to tune. This paper proposes an algorithm for automatically tuning a learning-rate schedule. The method works by modelling the training dynamics and adapting the learning-rate to optimize performance on the validation set. The method does end up introducing additional hyperparameters,
but the empirical results suggest some degree of robustness to them. Empirical performance is investigated on a number of widely-used models which are expensive to tune.

**Overall, I recommend this paper for acceptance.** I have some minor concerns which are detailed below, but generally the clarity was excellent throughout and the approach seems novel.

It seems that there are various "hidden" hyperparameters in addition to the ones stated. The algorithm seems to perform well in a variety of problems with them fixed, but there is a concern that the default settings suggested in the paper could just happen to fit well in these particular problems. For example,
  - **Page 4**: Don't you need to know (or guess) the variance of the noise term $\varepsilon$ to fit a GP? If so, how was this noise variance chosen? what's the sensitivity to it?
  - **Page 14, Appendix A.2**: *"Hence, we apply a simple gradient descent method that starts from an initial b, computes the linear least squares w.r.t. a, c under b, search for the next b by the gradient descent method, and repeats these two steps."*
    The problem in a,c can be solved in closed form but how is the step-size for descent in b chosen? Is this a tuned hyper-parameter? How was it chosen?
  - **Page 15, Appendix A.3**: *"At the end of each iteration, we remove the loss values that are among the farthest 3%"* How was 3\% chosen? This one in particular seems like it would depend a lot on how noisy the particular data is

My other concern is that there's no indication of statistical spread in any of results (neither tables nor figures). It  would be nice to see at least standard error bars so we know the variance in performance of the proposed method, and how it compares to the baselines. I assume this is because the experiments presented are too expensive to perform enough trials to get statistically significant results, but it would be worth-while to add an appendix showing results on smaller tasks on which variance can be assessed.

## Clarifications
---
- **Page 2**: *"Moreover, since it directly minimizes the validation loss, it does not only accelerate the convergence but also improves the generalization."* This isn't true necessarily, one could just be overfitting to the validation set. This is why we tune on a validation set and test performance on a separate test set --- to ensure that the performance truly does generalize to unseen data

## Minor Comments (which did not affect my score)
---
- **Page 4**: "loss decreases exponentially" vs. "linear convergence" (iteration complexity) is an unfortunate quirk in the optimization literature, as the equivalence is easy to miss. It would be useful for many readers if the equivalence was clarified --- in a footnote, perhaps --- or if the same convention was used in both phrases.
- I think an interesting addition to the empirical results would be something like "computation to performance" for each of the methods. For example, how much compute is it going to cost me to reach a certain level of performance using each method when we account for time spent tuning hyperparameters. I am unsure what the best measure would be here, but tuning non-trivial models is often so expensive overall that it would really showcase the benefits of the proposed method. (To be clear, *I am not requesting such a result*)

---

> ### Author Response · Authors · 2020-11-18
> **Response to Reviewer 3**
>
> We would like to thank you for your interest in our paper, and for thinking our approach is novel. We want to answer your questions and address your concerns here:
>
> - \> “Don't you need to know (or guess) the variance of the noise term ε to fit a GP? If so, how was this noise variance chosen? what's the sensitivity to it?”\
> We don’t need to know the variance of the noise term $\epsilon$ since the variance is part of the distribution model of $f(\eta)$ and BO automatically learns it (its estimation is the $\sigma_i(\eta)$ of the posterior distribution in Eq. 7). Please find a detailed definition of $\epsilon$ above Eq. 5: we do need to know the prior of $\epsilon$ and it is set as white noise (drawn from the normal distribution) in GP.
>
> - \> “The problem in a,c can be solved in closed form but how is the step-size for descent in b chosen? Is this a tuned hyper-parameter? How was it chosen?”\
> We use the conjugate gradient method. The step size is determined by the algorithm itself and it is optimal so we do not need to set up a step size.
>
> - \> “How was 3% chosen? This one in particular seems like it would depend a lot on how noisy the particular data is.”\
> Empirically, we observe that the loss curve can be noisy during the first 25% steps (in the worst case) as shown in Figure 5, and if we remove 3% per iteration for 10 iterations, we will have (97%)$^{10} \approx$ 75\% left, which matches the rest 75% steps without noise. In theory, this percentage can be infinitely small as long as we increase the iterations of doing the spline smoothing. We choose 3% because using a percentage too high may remove stable data points in early iterations. We did not heavily tune the percentage, but it works stably well on all of the DNN models in our experiments.
>
> - \> “There's no indication of statistical spread in any of the results.”\
> We did not include statistical spread because the experiments on large models are too expensive to perform with enough trials to get statistically significant results. However, we did report the performance of AutoLRS across multiple trials for training BERT-BASE (see Table 2 in A.5.2) and VGG-16 (see Table 5 in Appendix A.5.5). In this revision, we extended Table 6 by running two new groups of trials that train the VGG-16 model on CIFAR-10 for each $\tau_{\max}$, and report the mean and standard deviation results. To further measure the stability of AutoLRS, we did additional experiments to train VGG-16 with AutoLRS on CIFAR-100, and report the mean and standard deviation over 10 trials using different random seeds. The mean top-1 accuracy is 0.12\% higher than the hand-tuned LR schedule, and the best AutoLRS-generated LR schedule can achieve 0.37\% higher top-1 validation accuracy than the hand-tuned LR schedule. These results have been added to Appendix A.5.2 in the revision.
>
> - \> "Moreover, since it directly minimizes the validation loss, it does not only accelerate the convergence but also improves the generalization." This isn't true necessarily, one could just be overfitting to the validation set.\
> We changed the sentence to “Moreover, since it directly minimizes the validation loss, it does not only accelerate the convergence but also improves the generalization compared to just minimizing the training loss".
>
>
> Thank you for your review! Please let us know if you have any concerns or questions.

---

> > ### Comment · AnonReviewer3 · 2020-11-23
> > **Follow-up**
> >
> > Thanks for the detailed response. I've kept my score unchanged.
> >
> > > we do need to know the prior of and it is set as white noise (drawn from the normal distribution) in GP.
> >
> > This is what I was referring to; where do you get that prior? it seems like it couldn't be *known* because it is related to a modelling assumption rather than being a real property of the data. This suggests that it's a hyper parameter, in that you had to choose this prior. In particular you mention using a gaussian prior, but for this prior you would need to choose a variance; how was this prior variance chosen?

---

> > > ### Author Response · Authors · 2020-11-23
> > > **Response to the Follow-up**
> > >
> > > Thanks for clarifying your question!
> > >
> > > We used standard normal distribution $\mathcal N(0,1)$ as the prior. It has been widely adopted as the default prior in many previous BO approaches and experiments. In the Bayesian view, the BO performance is not sensitive to the choice of hyperparameters in the prior, and we also achieved robust performance across different experiments using this default prior.

---

### Official Review · AnonReviewer2 · 2020-10-28
**An automatic learning rate adjustment method based on Bayesian Optimization**

**Rating:** 6
**Confidence:** 2

**Review:**

The auto method presented in the paper is based on Bayesian optimization, but with an modification to make it less expensive. The method should be interesting to machine learning community.

The is clearly written and relatively easy to understand. The idea proposed in new and the experimental results demonstrate its effectiveness.

Pros:
1. The method is new and seems to be effective in speedup  the training several types of models. Using a forecast model to reduce running time is an interesting idea, thought the forecast model is an exponential time-series forecasting model.
2. The paper demonstrates the effectiveness of this method by pretty solid experiments. The three models used are popular and important ones.
3. The paper also describe two practical improvements( Gradually increase $\tau$ over the course of training, and Minimizing training loss in early stage)

Cons:
1. The method itself introduced a few hyperparameters including k, $\tau$, etc. Tuning them seems a manual process. The paper can say a bit more how and why choose values used in the experiment
2. The paper focuses on learning rate. In practice, there are many other hyperparameters that have impact on performance. How to make sure the results reported in paper is independent of the setting of other hyperparameters?

---

> ### Author Response · Authors · 2020-11-18
> **Response to Reviewer 2**
>
> We would like to thank you for sharing the same thinking with us that this method would be of great interest to the machine learning community. We want to address your two concerns here:
>
> 1.  \> “The method itself introduced a few hyperparameters.”\
> Yes, our algorithm has its own hyperparameters but we did not heavily tune them due to the expensive costs of the presented large-scale experiments. Although our default hyperparameter setting of AutoLRS in the paper may not be optimal, as shown in Section 4.3, it performs well for a diverse set of widely-used and state-of-the-art DNN models (ResNet, VGG, Transformer, and Bert) for CV and NLP tasks. We chose k = 10 and $\tau'=\tau/10$ because they strike an effective balance between the steps invested in search and the steps used for training, and this significantly saves computation time without severe performance degradation and thus makes our approach ready-to-use for ML practitioners. We agree that some of the hyperparameters are task-dependent, and fine-tuning them may lead to further improvements. The sensitivity test in Appendix A.5.2 suggests the robustness of AutoLRS to $\tau_{\max}$. We also presented ablation studies on the exponential forecasting model. In our new version, we added comparison of BO versus random search and grid search in Appendix A.5.4. The chosen GP kernel function and the acquisition function are commonly used choices in many applications of BO, and we did not tune them specifically for our task. We are happy to do additional experiments if you request them.
>
> 2. \> “There are many other hyperparameters that have impact on performance.”\
> For “other hyperparameters”, do you mean the training hyperparameters such as momentum, weight decay, batch size, etc? Please let us know if we misunderstand it. In all of our experiments (with the only exception of BERT pre-training, in which we used a larger batch size than the original used one), we utilized exactly the same training hyperparameters as in their original papers and official implementations except the learning rate, which were automatically adjusted by AutoLRS. Since these models have been widely used in both industry and academia, we postulate that their hyperparameters have been adequately tuned to gain a competitive performance.
>
> Thank you for your review! Please let us know if you have any concerns or questions.

---

> > ### Comment · AnonReviewer2 · 2020-11-23
> > **Answers to my questions**
> >
> > Thanks the authors for the answers to my questions. My primary concerns are the extra hyperparameters introduced by the proposed method and the reliability of tuning learning rate alone. I believe my concerns are mitigated by the authors' responses. I still think batch size, decay rate, etc. will have big impact of performance, but given that other comparing studies also similar settings, I'm ok with fixing them for the purpose of fair comparison. I keep my score unchanged.

---

### Official Review · AnonReviewer5 · 2020-11-02

**Rating:** 5
**Confidence:** 4

**Review:**

The paper presents an algorithm (AutoLRS) to tune learning rate schedules based on Bayesian Optimization with Gaussian processes and loss curve forecasting with exponential functions. The authors conduct experiments on three large scales settings in computer vision and natural language processing.

Strengths:
- This is probably one of the first attempts to use BO to fit LR schedules on the fly.
- The exponential forecast model seems a simple but effective idea.
- Experiments are very large scale and show that particular instantiations of the proposed algorithm achieve noticeable speed-ups and can also improve final performances in some cases.

Weaknesses:
- The method has several hyperparameters, which potentially limits its significance and applicability.
- Unclear contribution of the BO part of the method.
- Comparisons with other methods and experimental protocol could be improved.

Comments:
- AutoLRS has several hyperparameters.  The authors mention four in the relative paragraph, but considering also Sec 4.4 I personally count at least another three. They argue that some standard choices work well across the three experimented settings, but some of them (e.g. the number of iterations) are clearly task-dependent. Besides, there are also several choices related to the BO part of the method that feel somewhat arbitrary to me (choice of kernel function for the GP/acquisition function). Would they work also in another domain/learning setting, or should they also be considered as hyperparameters of AutoLRS? The presence of all these configuration parameters may render the application of the proposed algorithm impractical (lowering the significance of this work).
- The paper reads overall fairly well, but I think it could have been organized better. Some information (e.g. description of training details and in-depth reports of the numerical results) could have been postponed to the appendix and, vice-versa, some content of the appendix e.g. comparison with other methods should have been included in the main text.
- There is no ablation of the BO part of the method.  I find this quite surprising and I would like to see this in the rebuttal. Specifically, looking at Figure 6 (last column), it seems that after 10 samples the points are more or less evenly spaced (in log scale). I wonder if the BO part of the method adds that much. One could simply pick 10 values on a grid (or sample randomly) and only run the exponential forecasting part of the method. This reminds me of [1]; which could be related.
- I have doubts about the comparisons with other methods. A) It is not clear to me how the authors searched for the parameters of the schedulers (CLR/SGDR), besides that they did >= 10 trials. I would argue that manual search is not a fair method in this setting  B) Within the hypergradient descent class, the authors only reported some results with the method of Baydin et al. but none with Donini et al. (which is a generalization of HD that also considers optimizing with a validation set). C) Runs with Hyperband use exponential decay. Depending on which functional form they use, sampling the decay rate uniformly in (0,1) may be unnecessarily punitive. E.g. if the schedule is $\eta_0 \delta^{t}$ then clearly I would expect that only values of $\delta$ close to 1 would yield decent results. These points can be clarified in the rebuttal phase.
- The authors should be very careful when using the word "optimal" and related. As they say, the method that they propose is a greedy one that performs a (very) partial stagewise optimization. Therefore it is not clear to me concerning what they consider the various quantities as `"optimal". I strongly suggest replacing the occurrences with words such as "tuned" or similar.

Minor comments:
- The experimental results do not report any kind of measure of uncertainty, for instance, related to the random seed. While this is understandable for the large scale experiments, I would have appreciated a set of smaller-scale experiments to consider this.
- Algorithm 1 does not use explicitly $\eta_{min}$ and $\eta_{max}$. To improve clarity, authors should specify that line 5 solves a *constrained* optimization problem over $\eta$, in the range $[\eta_{min}, \eta_{max}]$
- A small explanation of the “LR range test” would be useful, due to the fact it is used to calibrate 2 of the 6 hyperparameters of the proposed method.


References
[1]Zhang, Michael, et al. "Lookahead optimizer: k steps forward, 1 step back." Advances in Neural Information Processing Systems. 2019.

---

> ### Author Response · Authors · 2020-11-18
> **Response to Reviewer 5 (Part 1)**
>
> We would like to thank you for your thorough review and insightful feedback. We are glad that you find our method effective and our experiments are large-scale. We are committed to addressing your comments and concerns, including adding additional experiments. We believe that your suggestions will help us significantly improve the paper. We want to address your concerns here:
> - \> “AutoLRS has several hyperparameters.”\
> As described in Section 4.3, though not heavily tuned due to the large-scale of the experiments, our default hyperparameter setting of AutoLRS performs well for a diverse set of widely-used and state-of-the-art DNN models (ResNet, VGG, Transformer, and Bert) for CV and NLP tasks. We choose k = 10 and $\tau'=\tau/10$ because they strike an effective balance between the steps invested in search and the steps used for training, and this significantly saves computation time without severe performance degradation and thus makes our approach ready-to-use for ML practitioners. We agree that some of the hyperparameters are task-dependent, and fine-tuning them may lead to further improvements. The sensitivity test in Appendix A.5.2 suggests the robustness of AutoLRS to $\tau_{\max}$. We also presented ablation studies on the exponential forecasting model. In our new version, we added a comparison of BO versus random search and grid search in Appendix A.5.4. The choices of kernel function for the GP and the acquisition function are commonly used in many other applications of BO, and we did not tune them specifically for our task. We are happy to do additional experiments if you request them.
>
> - \> “The paper reads overall fairly well, but I think it could have been organized better.”\
> Thank you for your suggestion! We have added experimental comparison with the other methods to Section 5 in the revision.
>
> - \> “There is no ablation of the BO part of the method.”\
> We conducted an ablation study to confirm the effectiveness of BO in our algorithm, and added the results of the ablation study in Appendix A.5.4, Tables 7-8. Compared to the performance of AutoLRS with BO in Table 6, the highest top-1 accuracy achieved in 350 epochs by random search and grid search are 0.09\% and 0.24\% behind BO respectively on average.
>
> - A) \> “It is not clear to me how the authors searched for the parameters of the schedulers (CLR/SGDR), besides that they did >= 10 trials.”\
> For CLR and SGDR, we first followed the guidelines of setting hyperparameters in their paper, and then adjusted their hyperparameters according to the training dynamics observed in our random trials. We want to emphasize that their recommended hyperparameters (e.g., CLR suggests to set half the cycle length to 2 - 10 times the number of iterations in an epoch) do not work well on all of the DNN models in our experiments. As a result, we carefully tuned their hyperparameters based on what the training dynamics looked like in the previous runs, and tried improving their performance in the next run. We believe we thoroughly simulated how an experienced ML practitioner would tune the hyperparameters of these two LR schedulers.\
> B) \> “Within the hypergradient descent class, the authors only reported some results with the method of Baydin et al. but none with Donini et al.”\
> Thank you for the pointer! We have added the comparison with MARTHE (Donini et al.) to Section 5 and Appendix A.5.3 in the revision.\
> C) \> “Runs with Hyperband use exponential decay. Depending on which functional form they use, sampling the decay rate uniformly in (0,1) may be unnecessarily punitive.”\
> Thank you for pointing this out! When we ran Hyperband to search for an exponential decay LR schedule, we sampled the decay factor uniformly in (0,1). We want to clarify that the $t$ in the schedule $\eta_0 \delta^{t}$ of our experiment is epoch rather than iteration, i.e., we decayed the learning rate by $\delta$ every epoch instead of every iteration. We found that $\delta$ in the range of [0.7, 1.0] produced decent top-1 accuracy results, but the $\delta$ that gives the top three best results are in the range of [0.9, 1.0]. Therefore, we set the search space of $\delta$ to (0.9, 1.0), and ran the Hyperband experiment again. Its performance improved compared to before, but it still failed to match the hand-tuned LR schedule. We have updated the results in Section 5 and Appendix A.5.3 in the revision.
>
> - \> “The authors should be very careful when using the word "optimal" and related.” \
> Thank you for pointing this out! We have replaced the occurrences of "optimal" with other words in our revision.
>
> References\
> Donini, Michele, et al. "MARTHE: Scheduling the Learning Rate Via Online Hypergradients." Proceedings of the 29th International Joint Conference on Artificial Intelligence and the 17th Pacific Rim International Conference on Artificial Intelligence. 2020.
>
> (continuing in next comment)

---

> > ### Author Response · Authors · 2020-11-18
> > **Response to Reviewer 5 (Part 2)**
> >
> > (continuing from the previous response)
> >
> > - \> “The experimental results do not report any kind of measure of uncertainty.”\
> > We extended Table 6 in the revision by running two more trials of training the VGG-16 model on CIFAR-10 for each $\tau_{\max}$, and reported the mean and standard deviation results. For experiments on those large models, due to our current limited computational resources, we cannot run them for more trials to get the measure of uncertainty. However, to provide a more complete evaluation of the variance of AutoLRS, we did additional experiments of training VGG-16 with AutoLRS on CIFAR-100, and reported the mean and standard deviation over 10 trials using different random seeds. The mean top-1 accuracy is 0.12\% higher than the hand-tuned LR schedule, and the best AutoLRS-generated LR schedule can achieve 0.37\%  higher top-1 validation accuracy than the hand-tuned LR schedule. These results have been added to Appendix A.5.2 in the revision.
> >
> > - \> “Algorithm 1 does not use explicitly $\eta_{min}$ and $\eta_{max}$.”\
> > We have added the constrained optimization statement to Section 4.3.
> >
> > - \> “A small explanation of the “LR range test” would be useful.”\
> > Thanks for pointing out! We have added a brief explanation of the LR range test to Section 2.
> >
> > Thank you again for your detailed review! Please let us know if anything remains unclear and we are happy to address any additional concerns and questions you might have.

---

> > ### Comment · AnonReviewer5 · 2020-11-20
> > **Contribution of BO does not seem statistically significant**
> >
> > Dear authors,
> >
> > thank you very much for your reply. I appreciate the revisions, your clarifications and the additional experiments.
> > I must say, however, that I remain unconvinced on the contribution of the BO part to the overall performance of the method. In fact, I fail to see statistical significance from the results that you reported in A.5.4. For instance, if I am not mistaken, you have that for $\tau_{max}=4000$ AutoLRS achieves $94.01\pm0.13$, hence the mean of the GS result, $93.89$, is well within two standard deviations. On the contrary, the ablative results of BO without exponential models lead to a much clearer conclusion.
> >
> > I also appreciate the clarifications about the comparisons and the additional results provided. I agree with you that the default values of LR schedulers are not particularly meaningful, but I still think that a fairer and more complete comparison also considering the runtime of all the methods would strengthen the submission (this is a somewhat less important point than the previous one).
> >
> > In conclusion, I have decided to raise my score to weak reject but I still cannot recommend acceptance. Once again, I appreciate the idea of the exponential model which seems to be the main component responsible for the good experimental results.
> > I believe that the community would benefit from being exposed to this idea. However, I would say that the work would be much stronger and the contribution much clearer with the BO part occupying a much less central part (perhaps a suggested extension, presented with"preliminary result"). This would also lead to a method with fewer hyperparameters.

---

> > > ### Author Response · Authors · 2020-11-23
> > > **Contribution of BO and runtime comparison with the other methods**
> > >
> > > We appreciate you raising the score. In the following, we try our best to further address your two concerns here:
> > >
> > > - \> “Contribution of BO does not seem statistically significant.”\
> > > We agree that the result of training VGG-16 on CIFAR-10 is not statistically significant, and we believe this is mainly due to the easy task of classification on CIFAR-10. We have observed more improvements of BO on training larger models for harder tasks, so we performed additional experiments on CIFAR-100 (100 classes makes the task harder). The top-1 accuracy achieved by AutoLRS-BO-generated LR schedules over 10 trials has a mean of 73.05\% and a standard deviation of 0.14\%. In contrast, the top-1 accuracy achieved by AutoLRS-GS-generated LR schedules over 10 trials has a mean of 72.63\% and a standard deviation of 0.56\%. On CIFAR-100, the mean of GS accuracy 72.63\% is out of two standard deviations from the BO accuracy 73.05\%$\pm$0.14\%. We added this new result to Appendix A.5.4. We believe that BO can show more advantages over grid search on larger models and larger data sets.
> > >
> > > - \> “I still think that a fairer and more complete comparison also considering the runtime of all the methods would strengthen the submission.”\
> > > In the latest revision of the paper, we additionally report the Runtime in a column of Table 2 in Section 5. This column shows how long each method takes on one NVIDIA Titan RTX GPU to find a good LR schedule for training VGG-16 on CIFAR-10. AutoLRS reduces the runtime by 28.6$\times$, 10.3$\times$, 16.7$\times$ compared to Hypergradient, MARTHE, and Hyperband, respectively. More importantly, the LR schedule found by AutoLRS is of higher quality. The runtime of the Hypergradient and MARTHE is more costly because they are sensitive to the hyperparameters and thus one needs to try many hyperparameter combinations to find a decent LR schedule. The overhead of Hyperband is due to the fact that it needs to evaluate various configurations with different numbers of epochs. Even with successive halving (early stopping), it still yields a large overhead.

---

> > > > ### Comment · AnonReviewer5 · 2020-11-23
> > > > **Contribution of BO and runtime comparison with the other method**
> > > >
> > > > Thank you for your reply.
> > > >
> > > > The differences on CIFAR100 looks indeed slightly more evident. Reading better, however, I've noticed that you always report the validation score, correct (also on the main paper)? I wonder if the difference is still evident on the test set. Could you provide these results?
> > > >
> > > > I am not entirely sure of the reason why you expect the effect of BO to be much more evident in more difficult settings. However, I could expect that the difference may be more evident when the search interval gets larger, as BO might dedicate more trials to more promising subintervals. This is not a request for additional experiments, as I know it is a bit late for that, but rather just a comment.
> > > >
> > > > Regarding comparisons and times, referring to Tab. 2., I sincerely do not think it is particularly fair to include the time for searching for good configurations only for other methods. I believe you should also search for AutoLRS parameters within reasonable ranges (at least $k, \tau, \kappa$, you only have reported sensitivity study for $\tau_{max}$, correct?).
> > > > By doing so, you might even end up with finding better configurations for AutoLRS and "dispel possible doubts" about the values you consider as "defaults".

---

> > > > > ### Author Response · Authors · 2020-11-23
> > > > > **Clarification on accuracy/scores reported**
> > > > >
> > > > > Thank you for your reply!
> > > > > - \> “I've noticed that you always report the validation score, correct (also on the main paper)?”\
> > > > > Thanks for pointing this out! This is a place where we need to improve the clarity of presentation. In particular, the paper reported validation accuracy for Imagenet since ImageNet does not disclose the labels of its test set and it is common in previous works to report the validation set accuracy. In contrast, CIFAR10 and CIFAR100 only have official training/test splittings so the reported accuracies on them are actually evaluated on their official test sets. For Transformer, we used the official test set of WMT 2014 English-German dataset to assess the performance. For BERT, we used the official Dev set of the downstream tasks to report the scores, because the test set is hidden. We added a footnote to clarify this in the latest revision of the paper, and updated the texts and tables in the appendix to make it clearer. Note in all experiments, the training does not have access to any validation/test samples used in evaluating the reported accuracies.
> > > > >
> > > > > - \> “I could expect that the difference may be more evident when the search interval gets larger, as BO might dedicate more trials to more promising subintervals.”\
> > > > > Thanks for your suggestion! We will definitely give it a try in the future.
> > > > >
> > > > > - \> “Regarding comparisons and times, referring to Tab. 2., I sincerely do not think it is particularly fair to include the time for searching for good configurations only for other methods.”\
> > > > > We agree that the setting you suggested is another comparison setting, which allows hyperparameter search for every method under the same runtime budget, and the comparison is between their achieved best accuracies after the same runtime. However, the setting used in our paper is also fair and is actually more advantageous to the baselines than to our method, and here is why. Considering a setting that compares the runtime of all methods needing to reach a pre-defined validation/test accuracy (i.e., a finish line), which has been commonly adopted by public leaderboards like Stanford’s DAWNBench (Coleman et al., 2017). If we set the finish line as the accuracy achieved by AutoLRS without any hyperparameter tuning, then the other baselines need more runtime than the reported runtime to reach the finish line because after the reported runtime of hyperparameter search they still cannot reach AutoLRS’s accuracy, according to the experimental results.
> > > > > We think that the two comparison settings are both fair, but the latter setting is closer to actual application scenarios, in which we usually stop tuning once the achieved result is sufficiently better than previous results. So we chose it for runtime comparison.
> > > > >
> > > > > References\
> > > > > Cody Coleman, Deepak Narayanan, Daniel Kang, Tian Zhao, Jian Zhang, Luigi Nardi, Peter Bailis, Kunle Olukotun, Chris Ré, and Matei Zaharia. “DAWNBench: An end-to-end deep learning benchmark and competition.” Training, 100(101):102, 2017.

---

### Decision · Program_Chairs · 2021-01-07
**Final Decision**

**Decision:**

Accept (Poster)

**Comment:**

The paper has been actively discussed, both during and after the rebuttal phase. I am thankful for the active communication that took place between the authors and some of the reviewers.

The paper was, overall, found quite clear, with an interesting methodology (especially the introduction of a forecasting step) and a solid large-scale experimental evaluation. As a result, it is recommended for acceptance.

However, several concerns remained after the rebuttal phase and we strongly encourage the authors to try to improve the following aspects of their submission:
* Clarify as much as possible (notably in the light of the ablation studies further added in the paper) the importance & impact of the BO component (which cast some doubts among the some reviewers on its necessity to get good performance)
* Transparently discuss the choice of, _and the robustness with respect to_, the “hyper-hyperparameters” of the proposed method (e.g., k, tau, tau’, kappa, tau_max, mini-batch size of validation set,...). Such an in-depth discussion is essential to fully demonstrate the practical value of the method.